# A conserved Mcm4 motif is required for Mcm2-7 double-hexamer formation and origin DNA unwinding

Kanokwan Champasa[1†], Caitlin Blank[1,2†], Larry J Friedman[3], Jeff Gelles[3]*, Stephen P Bell[1]*

[1]Department of Biology, Howard Hughes Medical Institute, Massachusetts Institute of Technology, Cambridge, United States; [2]Microbiology Graduate Program, Massachusetts Institute of Technology, Cambridge, United States; [3]Department of Biochemistry, Brandeis University, Waltham, United States

**Abstract** Licensing of eukaryotic origins of replication requires DNA loading of two copies of the Mcm2-7 replicative helicase to form a head-to-head double-hexamer, ensuring activated helicases depart the origin bidirectionally. To understand the formation and importance of this double-hexamer, we identified mutations in a conserved and essential Mcm4 motif that permit loading of two Mcm2-7 complexes but are defective for double-hexamer formation. Single-molecule studies show mutant Mcm2-7 forms initial hexamer-hexamer interactions; however, the resulting complex is unstable. Kinetic analyses of wild-type and mutant Mcm2-7 reveal a limited time window for double-hexamer formation following second Mcm2-7 association, suggesting that this process is facilitated. Double-hexamer formation is required for extensive origin DNA unwinding but not initial DNA melting or recruitment of helicase-activation proteins (Cdc45, GINS, Mcm10). Our findings elucidate dynamic mechanisms of origin licensing, and identify the transition between initial DNA melting and extensive unwinding as the first initiation event requiring double-hexamer formation.

DOI: https://doi.org/10.7554/eLife.45538.001

*For correspondence:
gelles@brandeis.edu (JG);
spbell@mit.edu (SPB)

[†]These authors contributed equally to this work

Competing interests: The authors declare that no competing interests exist.

## Introduction

Initiation of DNA replication occurs at genomic sites called origins of replication. Each eukaryotic origin is licensed during G1 phase through loading of the replicative helicase. The ring-shaped hetero-hexameric Mcm2-7 helicase is loaded around origin DNA by three loading proteins - ORC, Cdc6, and Cdt1 (*Bell and Labib, 2016*). The resulting licensed origin has two Mcm2-7 complexes encircling the DNA in a head-to-head orientation with their N-terminal domains interacting to form a 'double hexamer' (*Abid Ali et al., 2016*; *Evrin et al., 2009*; *Li et al., 2015*; *Noguchi et al., 2017*; *Remus et al., 2009*). Because they are poised to leave the origin in opposite directions, the head-to-head orientation of the loaded double hexamer is the first step in the establishment of bidirectional replication initiation. In the subsequent S phase, a subset of these helicases interact with two helicase-activating proteins, Cdc45 and GINS to form the active replicative helicase known as Cdc45-Mcm2-7-GINS (CMG) complex. This complex is then activated by Mcm10, leading to DNA unwinding, replication fork assembly and the initiation of DNA synthesis (*Douglas et al., 2018*; *Lőoke et al., 2017*; *Yeeles et al., 2015*). Separation of helicase loading and activation into distinct cell-cycle stages ensures that no origin can initiate replication more than once per cell cycle (*Bell and Labib, 2016*).

Helicase loading requires a series of ordered association and dissociation events. ORC-DNA binding is followed by Cdc6 association and recruitment of a Mcm2-7-Cdt1 complex to form a transient

four-protein intermediate called the ORC-Cdc6-Cdt1-Mcm2-7 (OCCM) complex (*Randell et al., 2006*; *Sun et al., 2013*). The ordered release of Cdc6 and Cdt1 along with ATP hydrolysis by the Mcm2-7 complex results in closing of the first Mcm2-7 ring around the origin DNA (*Ticau et al., 2017*; *Ticau et al., 2015*). Two models have been proposed to explain the recruitment and loading of the second Mcm2-7 (*Yardimci and Walter, 2014*). A two-ORC model suggests that there are two independent rounds of Mcm2-7 loading involving two ORC-Cdc6 complexes. These two loaded Mcm2-7 complexes are proposed to subsequently come together through translocation on the double-stranded origin DNA (*Coster and Diffley, 2017*; *Frigola et al., 2013*). This model is supported by the in vivo requirement for two ORC-binding sites at origins of replication and mutations that suggest that both the first and second Mcm2-7 complexes interact with ORC during loading (*Coster and Diffley, 2017*; *Frigola et al., 2013*). A second model proposes a single ORC molecule loads both the first and second Mcm2-7 (*Ticau et al., 2015*), with interactions between the first and second Mcm2-7 complex playing an important role during loading of the second Mcm2-7. This model is supported by single-molecule analysis of helicase loading, which observed that one ORC molecule is sufficient for Mcm2-7 double-hexamer formation (*Ticau et al., 2015*).

The Mcm2-7 helicase is a two-tiered, ring-shaped complex comprised of six distinct but related subunits (*Abid Ali and Costa, 2016*). The C-terminal domain (CTD) of each subunit is composed of AAA+ ATPase and winged-helix domains. The folded N-terminal domain (NTD) of each subunit includes an oligonucleotide/oligosaccharide-binding- (OB-) and, in some cases, a Zn-finger fold. In addition, Mcm2, Mcm4 and Mcm6 include extensive unstructured N-terminal extensions (NTEs). Although unrelated to one another, these NTEs are a conserved feature of these Mcm subunits in all eukaryotic species examined (*Miller and Enemark, 2015*). Once loaded, the tight interaction between the two Mcm2-7 complexes in the double hexamer is mediated by the NTDs (*Li et al., 2015*; *Noguchi et al., 2017*). It is clear that the Mcm4 and Mcm6 NTEs are critical targets of DDK during helicase activation (*Deegan et al., 2016*; *Randell et al., 2010*; *Sheu and Stillman, 2010*). The NTEs are not observed in structures of the Mcm2-7 double hexamer on or off the DNA (*Abid Ali et al., 2017*; *Li et al., 2015*; *Noguchi et al., 2017*) suggesting that the NTEs do not form a well-ordered structure at the interface holding the two hexamers together. Despite this, the NTEs are well positioned to facilitate initial interactions between the Mcm2-7 complexes and their role during helicase loading has not been examined in detail.

Although the formation of the Mcm2-7 double hexamer is critical to ensure bidirectional replication, it is unclear when or whether this form of the helicase is required for initiation. A key tool to address the role of double-hexamer formation would be separation-of-function mutations that allow loading of individual Mcm2-7 helicases but interfere with formation of the double hexamer. Such mutations would allow an investigation of which steps, if any, during replication initiation require the double-hexamer form of Mcm2-7.

To investigate the formation and function of the Mcm2-7 double hexamer, we initially focused on the function of the MCM NTEs. We made Mcm2-7 complexes lacking individual NTEs and found that constructs with a deletion of the Mcm2 or Mcm4 NTE were defective in helicase loading. During these studies, we identified a highly-conserved and essential motif within the Mcm4 NTD that is required for stable Mcm2-7 double-hexamer formation. Using single-molecule FRET analysis, we showed that mutation of this motif allows Mcm2-7 DNA loading and initial double-hexamer interactions, but the resulting complexes quickly dissociate. Although the separated Mcm2-7 complexes remain on the DNA, they do not form a double hexamer again. For both the wild-type and mutant Mcm2-7 complexes, all double hexamers form in a limited period of time after the DNA association of the second Mcm2-7. Mcm2-7 with a mutation in the Mcm4 motif cannot support replication initiation in vitro. Interestingly, these complexes can associate with helicase-activating proteins and perform initial origin DNA melting but fail to transition to extensive DNA unwinding. Our observations identify a key motif required for stable Mcm2-7-Mcm2-7 interactions and identify the transition to extensive origin DNA unwinding as the first step in replication initiation that requires the double-hexamer form of the helicase.

## Results

### N-terminal deletions of Mcm2 and Mcm4 inhibit Mcm2-7 loading

To study the role of Mcm2, Mcm4 and Mcm6 N-terminal extensions (NTEs) in helicase loading, we created deletion mutations lacking these elements (*Figure 1A*). We used Quick2D (*Dosztányi et al., 2005*; *Jones, 1999*; *Zimmermann et al., 2018*) and the location of phosphorylation sites (*Randell et al., 2010*) to identify the N-terminal unstructured regions for each subunit (*Figure 1—*

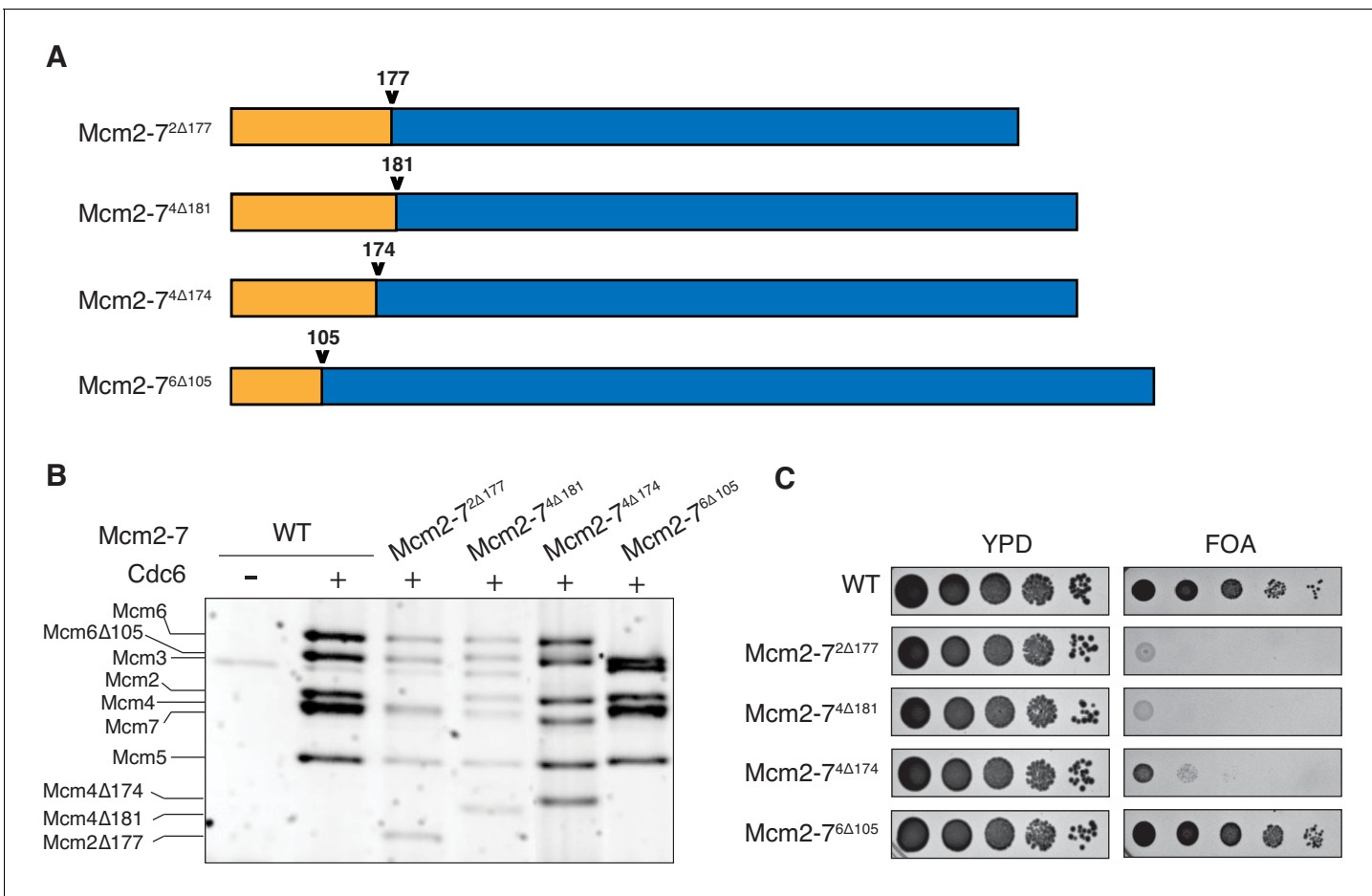

**Figure 1.** N-terminal deletions of Mcm2 and Mcm4 NTEs are defective for Mcm2-7 loading and essential for viability. (**A**) Diagram of Mcm2, Mcm4, and Mcm6 deletions. (**B**) Mcm2-7$^{2\Delta177}$ and Mcm2-7$^{4\Delta181}$ are defective in helicase loading onto origin-containing DNA. Helicase loading of the indicated Mcm2-7 N-terminal-deletions was monitored by incubating with purified helicase-loading proteins and bead-attached origin DNA followed by a high-salt wash (0.5M NaCl). All loading was dependent on Cdc6. (**C**) Mcm2Δ177 and Mcm4Δ181 deletions are lethal. The indicated genes were tested for complementation of a deletion of the corresponding wild-type gene before (YPD) and after (FOA) selecting against a plasmid with the corresponding wild-type gene. Endogenous *MCM2*, *MCM4* or *MCM6* gene is deleted, and a copy of the corresponding wild-type MCM is present on a *URA3*-containing plasmid. Indicated MCM mutants were integrated into the genome at the corresponding *TRP1* locus. Five-fold serial dilutions of cells were spotted on the indicated media.

DOI: https://doi.org/10.7554/eLife.45538.002

The following figure supplements are available for figure 1:

**Figure supplement 1.** Secondary structure prediction of Mcm2, 4, and 6 NTE by Quick2D, using the indicated analysis tools (PSIPRED, Protein secondary structure prediction based on position-specific scoring matrices as in *Jones, 1999*; IUPRED, the pairwise energy content estimated from amino acid composition discriminates between folded and intrinsically unstructured proteins as in *Dosztányi et al., 2005*; SS, alpha-helix/beta-strand; DO, Disorder).

DOI: https://doi.org/10.7554/eLife.45538.003

**Figure supplement 2.** Mcm2-7 complexes containing all wild-type or one mutant subunit (indicated) assembled into stable Mcm2-7 heterohexamers that bound to Cdt1.

DOI: https://doi.org/10.7554/eLife.45538.004

*figure supplement 1*). Based on these criteria, we purified Mcm2-7 complexes containing Mcm2$^{\Delta 2-177}$, Mcm4$^{\Delta 2-181}$, or Mcm6$^{\Delta 2-105}$. Subsequent structural studies (*Abid Ali et al., 2017*; *Li et al., 2015*; *Noguchi et al., 2017*) suggested that the first structured amino acids of Mcm2, Mcm4, and Mcm6 are amino acids 201, 177, and 103, respectively. Thus, we deleted the unstructured regions of Mcm2 and Mcm6, but included a small structured region in the Mcm4 NTE deletion. In addition, we purified Mcm2-7 containing Mcm4$^{\Delta 2-174}$, which bypasses the requirement of DDK in vivo (*Sheu and Stillman, 2010*) but had not been characterized biochemically. Importantly, upon expression with the other five wild-type subunits and Cdt1, each mutant Mcm subunit assembled into a stable Mcm2-7 heterohexamer that bound to Cdt1 (*Figure 1—figure supplement 2*). We will refer to the mutant Mcm2-7 complexes by the mutated subunit and the site of the mutation (e.g. Mcm2-7 that contains Mcm4$^{\Delta 2-181}$ is Mcm2-7$^{4\Delta 181}$).

To determine the impact of each deletion on Mcm2-7 loading, we tested the mutant Mcm2-7 complexes in an in vitro helicase-loading assay (*Remus et al., 2009*). After incubating the Mcm2-7 complex with purified ORC, Cdc6, Cdt1 and DNA, we used a high-salt wash to release helicase-loading proteins and Mcm2-7 complexes that had not completed loading (*Donovan et al., 1997*; *Randell et al., 2006*). Comparison with wild-type Mcm2-7 shows that Mcm2-7$^{2\Delta 177}$ and Mcm2-7$^{4\Delta 181}$ have loading defects (*Figure 1B*). In contrast, Mcm2-7$^{4\Delta 174}$ and Mcm2-7$^{6\Delta 105}$ are loaded at similar levels as the wild-type protein. Thus, the Mcm2 and Mcm4 deletions interfere with stable helicase loading. To determine whether these deletions impact MCM function in vivo, we assessed the ability of each of the mutant MCM genes to complement a deletion of the corresponding wild-type gene. Although Mcm2-7$^{2\Delta 177}$ and Mcm2-7$^{4\Delta 181}$ show incomplete helicase-loading defects in vitro, each of these mutants is lethal in vivo (*Figure 1C*). In contrast, Mcm2-7$^{6\Delta 105}$ supports normal cell growth, and Mcm2-7$^{4\Delta 174}$ cells are viable but grow at a much slower rate. Thus, the Mcm2 and Mcm4 deletion mutants impact both helicase loading and cell viability.

## A conserved Mcm4 N-terminal motif is required for stable double-hexamer formation

Because the NTEs are located at the double-hexamer interface, we asked if the NTE-deletion mutations impacted the formation of this complex. Previous studies have shown that double-hexamer formation is not required for salt-stable helicase loading detected in the previous assays (*Ticau et al., 2015*). Thus, we tested the ability of each of the mutant complexes to form Mcm2-7 double hexamers using a previously described gel-filtration assay (*Figure 2A*; *Evrin et al., 2009*). For wild-type Mcm2-7, Mcm2-7$^{2\Delta 177,}$ Mcm2-7$^{4\Delta 174}$ and Mcm2-7$^{6\Delta 105}$, the majority of salt-resistant DNA associated Mcm2-7 complexes eluted as double hexamers (*Figure 2B*). In contrast, loaded Mcm2-7$^{4\Delta 181}$ is entirely in the form of single hexamers (*Figure 2B*), suggesting that this mutation inhibits double-hexamer formation or stability.

The different functionality of the two Mcm4 mutants (Mcm2-7$^{4\Delta 174}$, no loading defect, viable; Mcm2-7$^{4\Delta 181}$, loading defect and lethal) focused our attention on the region of Mcm4 only present in Mcm2-7$^{4\Delta 174}$ (amino acids 175–181). To test the importance of this region in the context of otherwise full-length Mcm4, we constructed both substitution and deletion mutations (*Figure 3A*). Mcm2-7 complexes containing these mutations assembled normally and interacted with Cdt1 (Figure 1-figure supplement 2). Deletion or alanine substitution of these amino acids causes a two-fold loading defect compared to wild type Mcm2-7 (*Figure 3A*). In the double-hexamer assay, loaded Mcm2-7$^{4\Delta 175-181}$ and Mcm2-7$^{4-7A}$ eluted almost exclusively in the form of single hexamers (*Figure 3B*), implicating this region of Mcm4 in double-hexamer formation or stability. Importantly, both *mcm4Δ175–181* and *mcm4-7A* are unable to complement a *MCM4* deletion (*Figure 3—figure supplement 1*).

Sequence analysis of the 175–181 region of Mcm4 identified a highly-conserved nine-amino-acid motif that overlapped with this region (*Figure 4A*). To test whether this motif is required for double-hexamer formation, we constructed a series of substitution mutations overlapping this motif (*Figure 4B*), incorporated them into otherwise wild-type Mcm2-7 complexes (*Figure 1—figure supplement 2*) and tested them for helicase loading and double-hexamer formation. Mcm2-7 complexes with mutations that disrupt the conserved motif (Mcm2-7$^{4-178A}$, Mcm2-7$^{4-181FA}$, Mcm2-7$^{4-182A}$ and Mcm2-7$^{4-185A}$) show two-fold loading defects compared to wild-type, whereas mutant complexes that do not alter the conserved regions (Mcm2-7$^{4-175A}$ and Mcm2-7$^{4-188A}$) show near-wild-type levels of helicase loading (*Figure 4C*). Testing these mutants for the formation of stable double

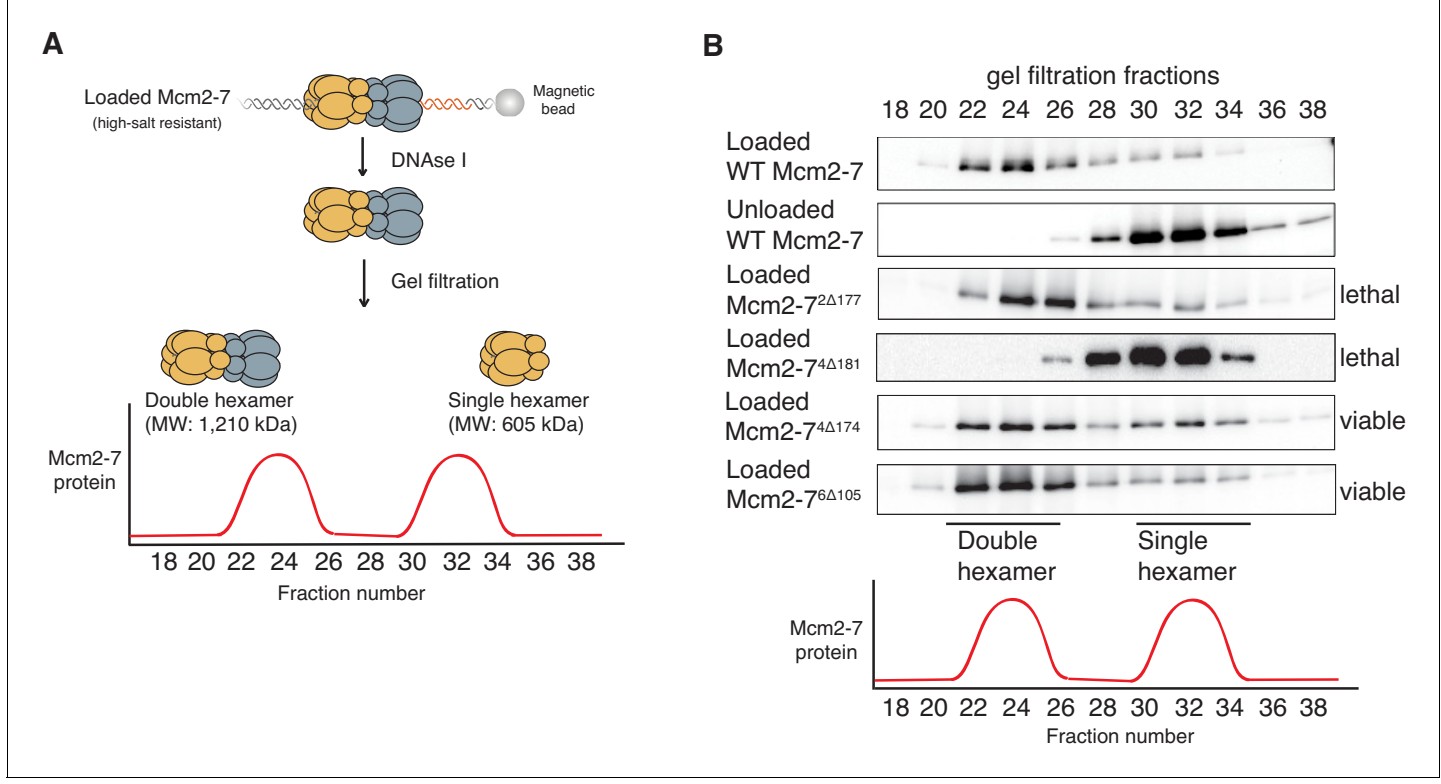

**Figure 2.** Deletion of the Mcm4 N-terminal region inhibits Mcm2-7 double-hexamer formation. (**A**) Double-hexamer assay scheme. After performing a helicase-loading reaction with purified proteins and high-salt wash, loaded Mcm2-7 complexes were released from the bead-bound DNA by DNase I treatment. Released Mcm2–7 complexes were fractionated by gel filtration to separate double from single hexamers. (**B**) Mcm2-7$^{4\Delta181}$ is defective for double-hexamer formation. Mcm2-7 complexes including the indicated N-terminal deletions were tested for their ability to form double hexamers as described in (**A**). The Flag-Mcm3 subunit of the Mcm2-7 complexes in the indicated fractions was detected by immunoblot. Wild-type Mcm2-7 double hexamers eluted early (fraction 22–26), whereas purified Mcm2-7 eluted later (fraction 30–34). Mutant viability data (right) are from *Figure 1C*.
DOI: https://doi.org/10.7554/eLife.45538.005

hexamers shows that Mcm2-7$^{4\text{-}178A}$, Mcm2-7$^{4\text{-}181FA}$, Mcm2-7$^{4\text{-}182A}$ were primarily in single-hexamer form and Mcm2-7$^{4\text{-}185A}$ is split between single- and double-hexamer forms (*Figure 4D*). In contrast, the mutants that are outside of the conserved motif are primarily in the double-hexamer form (*Figure 4D*). These results demonstrate that this conserved Mcm4 motif is important for the formation or stability of the Mcm2-7 double hexamer. Therefore, we named this region the Double-Hexamer Motif (DoHM). Consistent with double-hexamer formation being essential, we observed a close correlation between the ability of a mutant Mcm4 to form stable double hexamers and the ability of the corresponding mutant gene to complement a *MCM4* deletion (*Figure 4D*, *Figure 4— figure supplement 1*).

## The DoHM is required for Mcm2-7 double-hexamer stability

The DoHM mutants could result in reduced double-hexamer detection for two possible reasons. The first possibility is that these Mcm4 mutants allow double-hexamer formation, but the resulting assemblies fall apart due to a weakened interface. Alternatively, the DoHM could be required to initiate double-hexamer formation. In the experiments described thus far, we used an ensemble assay to monitor double-hexamer formation. This assay requires the two Mcm2-7 complexes to remain associated during size-exclusion chromatography to be detected. Thus, defects either in the rate of stable double-hexamer formation or in the lifetime of the double-hexamer complex would not be distinguished by this assay.

To monitor double-hexamer interactions in real time, we used a single-molecule FRET assay for double-hexamer formation (*Ticau et al., 2015*). In this experiment, the N-terminus of the Mcm7 subunit in two separate Mcm2-7/Cdt1 preparations was labeled with either a donor or an acceptor

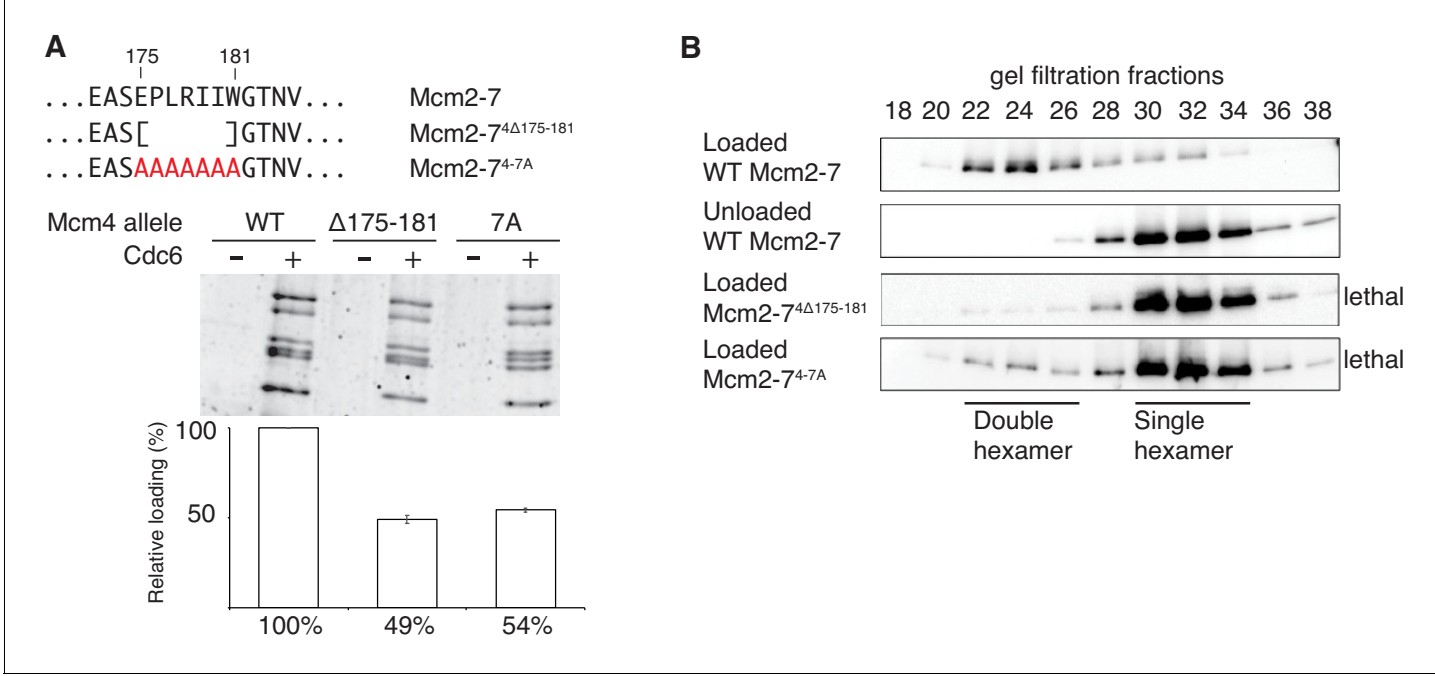

**Figure 3.** Amino acids 175–181 of Mcm4 are important for Mcm2-7 loading and double-hexamer formation. (**A**) *Top*, diagram of Mcm4 amino acids 175–181 deletion and alanine-substitution mutations. *Bottom*, Mcm2-7$^{4\ \Delta175\text{-}181}$ and Mcm2-7$^{4\text{-}7A}$ are defective for helicase loading. Helicase loading of wild-type or Mcm2-7 complexes including the indicated Mcm4 mutant was monitored as described in *Figure 1*. The associated graph shows the relative loading of the Mcm4 mutants compared to wild-type Mcm2–7 based on three independent experiments. Error bars indicate the SD. (**B**) Mcm2-7$^{4\Delta175\text{-}181}$ and Mcm2-7$^{4\text{-}7A}$ are defective in double-hexamer formation. Double-hexamer formation by Mcm2–7 complexes containing the indicated Mcm4 mutation was tested as described in *Figure 2*. Mutant viability data (right) are from *Figure 3—figure supplement 1*.

DOI: https://doi.org/10.7554/eLife.45538.006

The following source data and figure supplement are available for figure 3:

**Source data 1.** Primary data and quantification for graph in *Figure 3A*.

DOI: https://doi.org/10.7554/eLife.45538.008

**Figure supplement 1.** Elimination of the Mcm4 DoHM motif is lethal.

DOI: https://doi.org/10.7554/eLife.45538.007

fluorophore. We incubated surface-tethered fluorescent origin DNA with an equimolar mixture of the two differentially-labeled Mcm2-7 complexes together with the three helicase-loading proteins (ORC, Cdc6 and Cdt1, *Figure 5A*). We alternately excited the acceptor and donor fluorophores to detect the colocalization of donor- or acceptor-labeled Mcm2-7 with the fluorescently labeled DNA, a proxy for DNA binding. Importantly, when the donor fluorophore is excited, we are also able to detect hexamer-hexamer interactions through an increase in apparent FRET efficiency (EFRET, *Ticau et al., 2015*). To eliminate helicase-loading events during which we cannot monitor these interactions, we only analyzed events in which we observed DNA association of one donor-fluorophore-labeled and one acceptor-fluorophore-labeled Mcm2-7 complex, and measured $E_{FRET}$ only after arrival of the second Mcm2-7.

Single-molecule analyses of double-hexamer formation supported a role for the DoHM in double-hexamer formation. Both wild-type Mcm2-7 and a DoHM mutant (Mcm2-7$^{4\text{-}178A}$, referred to as the DoHM mutant hereafter) exhibited an increase in $E_{FRET}$ shortly (<10 s) after the arrival of the second Mcm2-7 (*Figure 5B,C*; *Figure 5—figure supplement 1*). Analysis of wild-type helicase-loading events (N = 89) revealed that immediately after arrival of the second Mcm2-7, the two-Mcm2-7 complexes were primarily in state that exhibited zero or very low FRET ($E_{FRET}$ = 0.029 ± 0.003, *Figure 5D–i and D-ii*, *Supplementary file 1 - table 1*). However, shortly after the second Mcm2-7 arrived, many (~70%; *Supplementary file 1 - table 1*) of the DNAs associated with wild-type Mcm2-7 transitioned to a state exhibiting increased FRET ($E_{FRET}$ = 0.606 ± 0.002; *Figure 5D–iii and D-iv*, *Supplementary file 1 - table 1*). Importantly, all the complexes that transitioned to high $E_{FRET}$

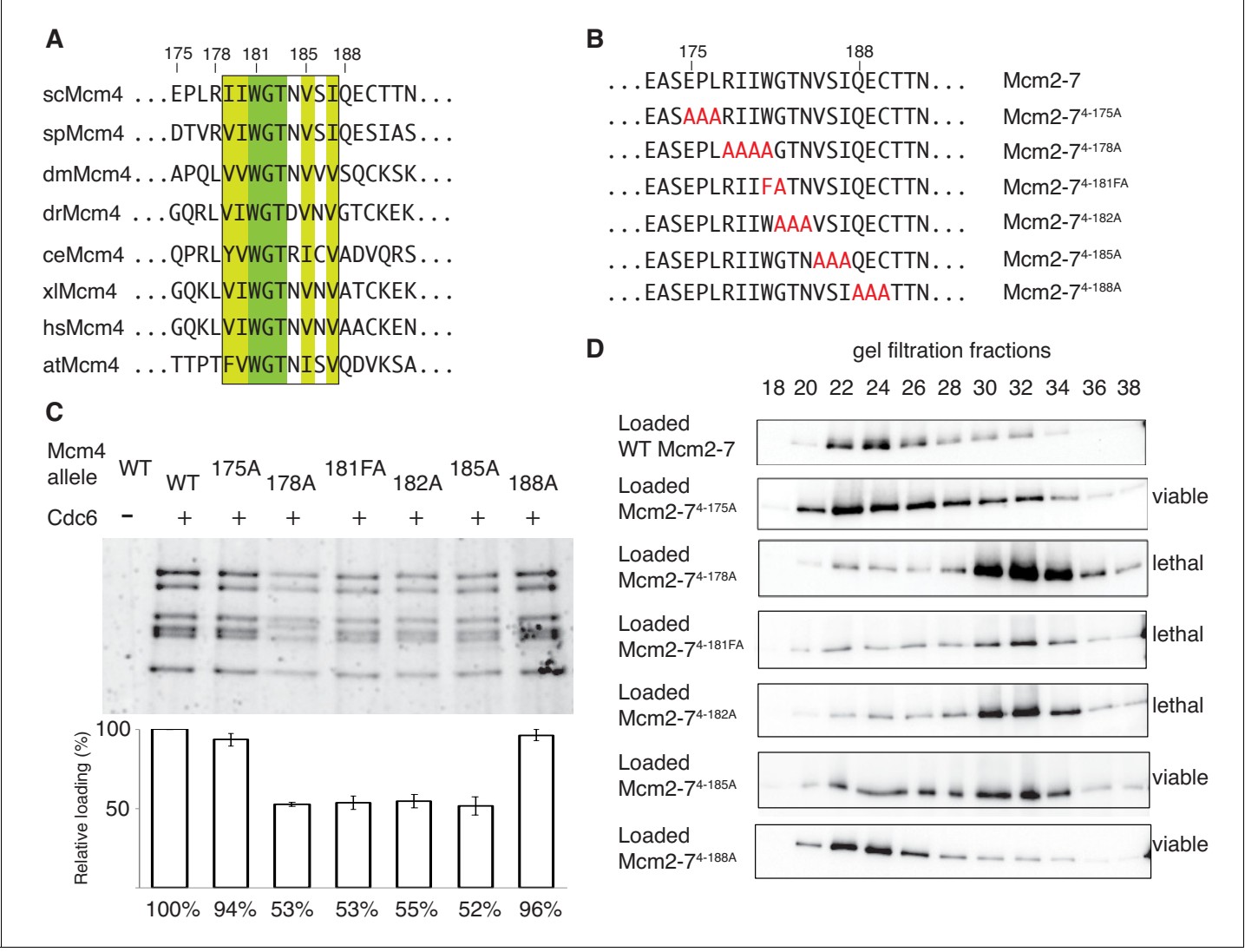

**Figure 4.** The DoHM is an essential and conserved region within Mcm4 that is required for Mcm2-7 double-hexamer formation. (A) Protein-sequence comparison of amino acids (175 to 194) surrounding the of *S. cerevisiae* DoHM (boxed) with Mcm4 of the indicated species (sp, *S. pombe*; dm. *D. melanogaster*; dr, *D. rerio*; ce, *C. elegans*; xl, *X. laevis*; hs, *H. Sapien*; at, *A. thaliana*). Identical amino acids are shaded dark green, and conserved amino acids are shaded light green. (B) Diagram of alanine mutations spanning the DoHM and the flanking amino acids. (C) Mcm4 mutations within the DoHM show helicase-loading defects. Helicase loading of Mcm2–7 complexes containing indicated Mcm4 mutants was monitored as described in *Figure 1*. The associated graph shows the relative loading of the Mcm4 mutants compared to wild-type Mcm2-7, based on three independent loading experiments. Error bars indicate the SD. (D) Mcm4 mutations within the DoHM are defective for double-hexamer formation. Double-hexamer formation by Mcm2-7 complexes containing indicated Mcm4 mutant was tested as described in *Figure 2*. Mutant viability data are indicated to the right and are from *Figure 4—figure supplement 1*.

DOI: https://doi.org/10.7554/eLife.45538.009

The following source data and figure supplements are available for figure 4:

**Source data 1.** Primary data and quantification for graph in *Figure 4C*.
DOI: https://doi.org/10.7554/eLife.45538.012
**Figure supplement 1.** Mutation of the DoHM is lethal.
DOI: https://doi.org/10.7554/eLife.45538.010
**Figure supplement 2.** Location of the Mcm4 DoHM in the context of the Mcm2-7 double hexamer bound to DNA (*Abid Ali et al., 2017*).
DOI: https://doi.org/10.7554/eLife.45538.011

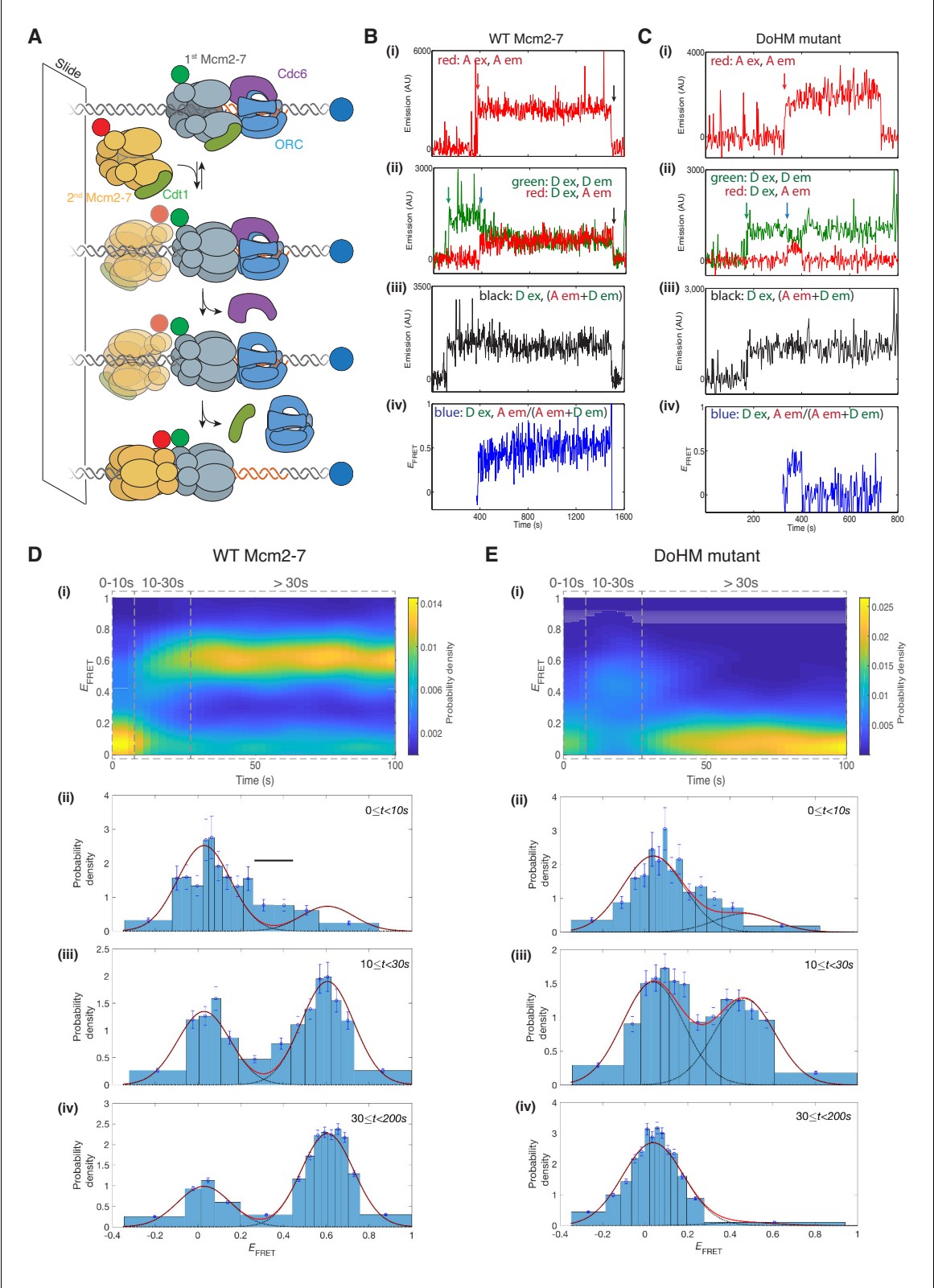

**Figure 5.** The DoHM is required for double-hexamer formation and stability. (**A**) Schematic of the single-molecule helicase-loading assay. Alexa-Fluor-488-labeled (blue circle) 1.3 kb origin DNAs were coupled to a passivated microscope slide. Mcm2-7 was fluorescently labeled with donor (Dyomic-549, green circle) or acceptor (Dyomic-649, red circle) fluorophores. Purified ORC, Cdc6, and Cdt1/Mcm2–7 were incubated with slide-coupled DNA. Colocalization of fluorescently-labeled proteins with the DNA and any associated FRET signal were monitored. (**B**) Wild-type Mcm2-7 forms long-lasting

*Figure 5 continued on next page*

*Figure 5 continued*

FRET signals. Representative fluorescence records for an experiment using a 1:1 mixture of wild-type donor- and acceptor-labeled Mcm2-7 showed FRET after arrival of the second Mcm2–7. Records of fluorescence intensity for (i) acceptor excitation; acceptor emission (Dyomic-649-labeled Mcm2-7, red arrow marks arrival of acceptor-labeled Mcm2-7), (ii) donor excitation; donor emission (Dyomic-549-labeled Mcm2-7, green arrow marks arrival of donor-labeled Mcm2-7) and FRET (donor excitation; acceptor emission, blue arrow marks initiation of FRET), (iii) total emission (donor excitation; donor emission + acceptor emission), and (iv) calculated $E_{FRET}$ are shown. Black arrows indicate both donor and acceptor release due to the double hexamer sliding off the end of the DNA. (C) Representative fluorescence records for experiments using Mcm2-7[4-178A] (labels and arrows as in B) show a short-lived FRET signal. (D) (i) Time evolution of the $E_{FRET}$ distribution for 89 wild-type Mcm2-7 complexes. Only complexes with one donor-labeled and one acceptor-labeled Mcm2-7 were selected. $E_{FRET}$ values were measured only after arrival of the second Mcm2-7, which was taken to be time zero in each record. The plot is a two-dimensional histogram (see Materials and Methods) with $N_t = 2688$ measurements within the time and $E_{FRET}$ range shown. (ii-iv) Histograms (probability density ± S.E.) of $E_{FRET}$ values recorded during the indicated time intervals after association of the second Mcm2–7 with origin DNA (black bar indicates possible intermediate $E_{FRET}$ state). $E_{FRET}$ values were globally fit to the sum (dashed cyan curves) of two Gaussians (red curves) constrained to have the same peak positions and widths at all times. (E) Analyses analogous to (D) for the DoHM mutant (114 complexes; $N_t = 2584$ in range shown in (i). For (ii) - (iv) in (D, E), fit parameters and numbers of observations are reported in ***Supplementary file 1 - table 1***.
DOI: https://doi.org/10.7554/eLife.45538.013
The following figure supplement is available for figure 5:

**Figure supplement 1.** Additional fluorescence records for experiments using a 1:1 mixture of donor- and acceptor-labeled wild type Mcm2-7 (A) or the DoHM mutant (B) showed FRET after arrival of the second Mcm2–7.
DOI: https://doi.org/10.7554/eLife.45538.014

remained in that state (***Figure 5D–i and D-iv***) until the observation was terminated by the end of the recording or by the disappearance of donor and/or acceptor fluorescence (either due to photobleaching or sliding of the double hexamer off the DNA; ***Figure 5B***, black arrow). These observations are consistent with our previous studies showing that wild-type Mcm2-7 rapidly forms stable double hexamers upon recruitment of the second Mcm2-7 (***Ticau et al., 2015***). In the data collected shortly (~0–10 s) after arrival of the second Mcm2-7, there is some evidence for the presence of a transient state with $E_{FRET}$ intermediate between zero and the high value (***Figure 5D–ii***, black bar) but the additional state, if present, was not clearly resolved in the current experiments.

After arrival of the second Mcm2-7, the DoHM mutant also exhibited an initial state with similarly low FRET followed by a rapid transition on many of the DNA molecules to a state with high FRET (***Figure 5C***). Thus, like wild-type Mcm2-7, the DoHM mutant can form a structure in which the two hexamers are so closely apposed that the donor and acceptor fluorophores are separated by only a few nanometers. However, there are two noticeable differences in the $E_{FRET}$ time courses with the DoHM mutant helicase (***Figure 5E***). First, the elevated $E_{FRET}$ value observed for the DoHM mutant was distinct (0.468 ± 0.010) and significantly lower than that observed for wild-type Mcm2-7 (0.606 ± 0.002, ***Supplementary file 1 - table 1***). This difference suggests that the high $E_{FRET}$ state of the DoHM mutant has a three-dimensional structure and/or dynamics that differ from those of the wild-type double hexamers. Second, the DoHM mutant complexes that transition to the elevated $E_{FRET}$ state do not remain there. Instead, the helicases quickly (in ~20 s) transition back to a state in which both hexamers are still bound to the DNA but which display low FRET (***Figure 5E–i and E–iv***). Together, these data indicate that the double-hexamer interactions formed by the DoHM mutant are incomplete and unstable.

## Mcm2-7-Mcm2-7 interactions occur rapidly or not at all

To examine the formation and stability of the double-hexamer complexes, we analyzed all DNA molecules associated with one donor- and one acceptor-labeled Mcm2-7 to determine the timing of transitions between the low-$E_{FRET}$ and the high-$E_{FRET}$ states observed in the wild-type and the DoHM mutant FRET records. Both of these data sets contained significant subpopulations of DNA molecules that have bound both donor- and acceptor-labeled Mcm2-7 but which do not exhibit FRET at any point in the recording. This suggests that some complexes exist in a configuration that is refractory to double-hexamer formation, a configuration that we will refer to as non-double hexamer (non-DH). Therefore, a minimal kinetic scheme to explain the wild-type Mcm2-7 data contains at least three molecular states: the high $E_{FRET} = ~0.45–0.6$ state that we interpret as double-hexamer (DH) and two states with indistinguishable $E_{FRET} = ~0$ values, but that can be distinguished by their lifetimes: the long-lived non-DH state and the transient pre-double-hexamer (pre-DH) state that

exists in the short time interval between binding of the second Mcm2-7 and the formation of a DH (see *Figure 5D*). We globally fit our set of wild-type Mcm2-7 FRET data to a kinetic model containing these three states (*Figure 6A*).

The fitting algorithm (see Materials and methods) yielded estimates for all six possible first-order rate constants connecting the three states (*Figure 6A*; *Figure 6—figure supplement 1*) and also assigns one of the three states to each molecule at every time point in the record (*Figure 6C*) based on state duration and $E_{FRET}$.

The kinetic modeling for wild-type Mcm2-7 reveals that 63% (52 of 82) of second Mcm2-7 binding events resulted in double-hexamer formation, all of which occur within thirty seconds of the second Mcm2-7 arrival. In most cases (41 of 52), these molecules start in the pre-DH state and rapidly convert to the DH state with a rate constant of $0.150 \pm 0.075$ s$^{-1}$ (*Figure 6A*). The rate constant for the reverse transition from DH to pre-DH is not significantly different from zero (*Figure 6—figure supplement 1*), suggesting that the pre-DH to DH transition is essentially irreversible. These data are consistent with the interpretation that the wild-type Mcm2-7 double hexamer is stable once formed (*Figures 5B* and *6A*, and *Figure 5—figure supplement 1*; *Abid Ali et al., 2017*; *Li et al., 2015*; *Noguchi et al., 2017*). Fourteen percent (11 of 82) of molecules were assigned to already be in DH state at the first time point. These events are likely to represent molecules whose sojourn in pre-DH state was too short to be detected in an experiment with this time resolution (~2.67 s). In contrast to the transition between pre-DH and DH, we do not detect significant non-zero rate constants for entering or leaving the non-DH state, consistent with the interpretation that the 35% of molecules (29 of 82) assigned to non-DH at the first time point are a refractory population trapped in a dead-end state (see Discussion). Together, these data indicate that once two wild-type Mcm2-7 complexes associate with DNA they either rapidly transition to a double hexamer or remain as a pair of single hexamers.

We also analyzed the Mcm2-7$^{4-178A}$ $E_{FRET}$ data by fitting it to a three-state model analogous to that used for the wild-type data. Three characteristics of the resulting analysis are similar between wild-type and the DoHM mutant (*Figure 6B*). First, in most cases, the arrival of the second Mcm2-7 was followed by a rapid transition from an initial very low-FRET state (pre-DH) to a higher $E_{FRET}$ state (62 of 100) or the high-FRET state was present at the first time point (13 of 100). Second, these transitions occurred rapidly and with a similar rate constant ($0.13 \pm 0.04$ s$^{-1}$) to wild-type ($0.15 \pm 0.08$). Third, the remaining second Mcm2-7 binding events (25 of 100) never exhibited the high-FRET state and were categorized as in the non-DH state. Thus, like the wild-type helicase, the DoHM mutant either rapidly exhibited a high-FRET state or remained in the low-FRET state throughout the observations.

There were two significant differences in the kinetics of double-hexamer interactions exhibited by the DoHM mutant helicase. As mentioned earlier, the high $E_{FRET}$ state of the mutant is different from that of the DH state of the wild type ($E_{FRET}^{DoHM} = 0.468 \pm 0.010$; $E_{FRET}^{WT} = 0.606 \pm 0.002$; *Figure 5D,E* and *Figure 6—figure supplement 1*). Thus, we refer to this state as the pseudo-double hexamer (pseudo-DH). A second distinction is the short-lived nature of this state. The DoHM mutant rapidly ($0.052 \pm 0.007$ s$^{-1}$) transitioned from the pseudo-DH state back to a low-FRET state (*Figure 6B*). Interestingly, the state that follows the pseudo-DH is a long-lived low-FRET state, which we termed post-double-hexamer (post-DH). This state is kinetically distinct from the short-lived low-FRET pre-DH state (*Figure 6D*) but is indistinguishable from the non-DH state; both are long-lived and exhibit similar low-FRET values. Thus, these states are grouped together into a single non-DH/post-DH state for the kinetic analysis (*Figure 6B,D*). Although the mutant pre-DH molecules rapidly transitioned to pseudo-DH and then to post-DH, the rates of the reverse reactions (pseudo-DH to pre-DH and post-DH to pseudo-DH) were both calculated to be > 100 times slower and not significantly non-zero (*Figure 6—figure supplement 1*). These findings suggest that both of these transitions are essentially irreversible and that once the two mutant Mcm2-7 complexes that form a pseudo-DH separate they will not form subsequent stable interactions. Records that were interpreted as showing a transition from non-DH/post-DH back to the high $E_{FRET}$ pseudo-DH state were rare (3 of 100) and sometimes showed only a single time point in the pseudo-DH state (*Figure 6D*, e.g. trajectory 67), possibly arising from infrequent errors in state assignment. Even though most or all post-DH molecules never returned to a high $E_{FRET}$ state, we find that both the DoHM mutant complexes remain associated with the DNA for extended times (*Figure 6D*, *Figure 5—figure supplement 1*), consistent with both helicases having successfully encircled the DNA. These findings

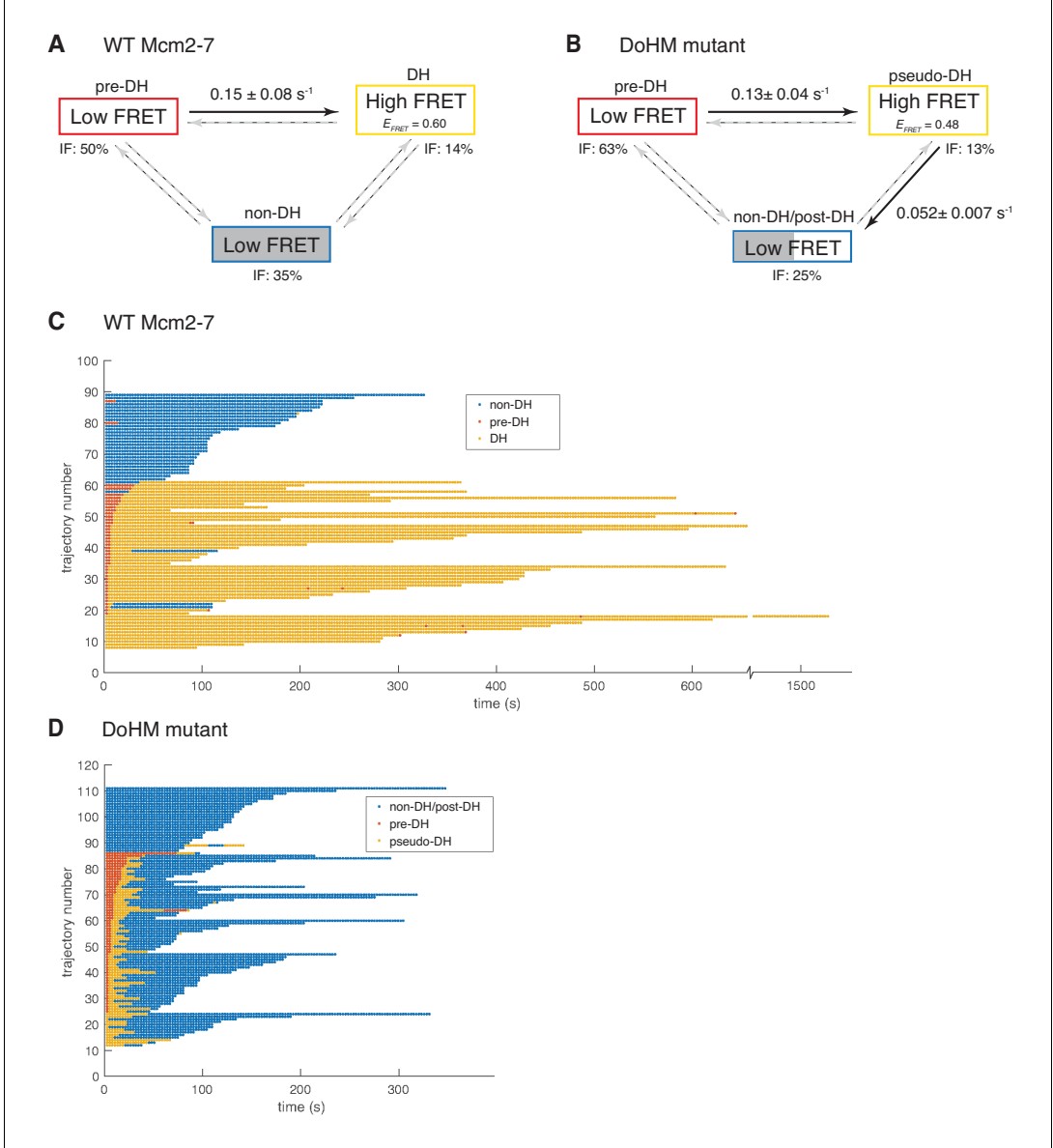

**Figure 6.** Wild-type Mcm2-7 and the DoHM mutant attempt double-hexamer formation only once. To focus on double-hexamer formation, analysis was restricted to the time interval from second Mcm2-7 binding to the time of one or both Mcm2-7 departure. (A–B) Kinetic models of transition among the three indicated $E_{FRET}$ states. Pooled single-molecule fluorescence records from helicase loading experiments using (A) Mcm2-7 or (B) Mcm2-7[4-178A] were globally fit (see Materials and methods) to generalized three-state kinetic models (i.e. all possible inter-state transitions were allowed). Black arrows indicate kinetically significant reaction steps; gray arrows indicate steps with rate constants that are not significantly greater than zero (see *Figure 6—figure supplement 1*). Fits also yielded the fraction (IF) of molecules in each state at the time of second hexamer binding. In the wild-type Mcm2-7 model (A), the transition from pre-DH to DH is the only process that occurs at an appreciable rate, while in the DoHM-mutant model (B) both pre-DH to pseudo-DH and pseudo-DH to post-DH steps are significant. (C–D) State rastergrams from the kinetic models. Each trajectory (horizontal line) indicates the state assigned by the model to a single-molecule record with wild-type Mcm2-7 (C) or the DOHM mutant (D), starting from the time of second hexamer binding. Pre-DH (red lines), DH or pseudo-DH (yellow lines) and post-DH or non-DH (blue lines) states are indicated. Trajectories are sorted by the onset time of the DH or pseudo-DH state, then by record length. Each record ends either due to the end of the experiment or loss of signal of one or both Mcm2-7 molecules. The blank region at the bottom of each rastergram represents low signal-to-noise records excluded from analysis (see Materials and Methods). Rare transitions (e.g. occasional transitions into or out of non-DH in wild-type Mcm2-7) which do not occur at statistically significant rates may reflect ambiguities in assigning time segments, particularly when the transitions are between two states with the same (zero) $E_{FRET}$.

DOI: https://doi.org/10.7554/eLife.45538.015

The following figure supplement is available for figure 6:

**Figure supplement 1.** Complete set rate constants (s$^{-1}$) and $E_{FRET}$ values derived from the kinetic model fits (*Figure 6*).

*Figure 6 continued on next page*

*Figure 6 continued*

DOI: https://doi.org/10.7554/eLife.45538.016

support a model in which there is a limited window of opportunity to form double-hexamer interactions during helicase loading after which two loaded Mcm2-7 complexes cannot interact.

We also noted that the lifetime of two single hexamers associated with the DNA is noticeably shorter than that of a double hexamer (*Figure 6C and D*, non-DH and post-DH states, blue lines, note that release of either of the two Mcm2-7 present results in the end of the record as we are looking for double-hexamer formation). Two wild-type Mcm2-7 complexes in the context of a double-hexamer can remain on the DNA for over 600 s, with many lasting to the end of observation (*Figure 6D*, yellow lines). In contrast, the retention of both of two single hexamers with the DNA whether they are wild-type or DoHM mutants is much shorter (~300 s, *Figure 6C and D*, blue lines). The average duration of DNA association for two wild-type Mcm2-7 complexes is 288 ± 23 s compared to 140 ± 9 s for the DoHM mutant. This observation suggests that the single Mcm2-7 complexes are less stably associated with the DNA relative to Mcm2-7 complexes in the form of double-hexamers. This difference is consistent with the reduced loading observed in the bulk experiments with the DoHM mutant.

## Double-hexamer formation is required for later steps of replication initiation

Our analysis of the Mcm2-7$^{4-178A}$ mutant indicates that the DoHM is important for double-hexamer formation and stability. Despite these defects, many of the mutant single hexamers remain stably bound to DNA. This allowed us to ask which, if any, of the subsequent steps in helicase activation and replication initiation can occur with only single-hexamers loaded onto DNA. To this end, we assessed the ability of the DoHM mutant to associate with helicase-activating proteins, unwind origin DNA and replicate DNA. Consistent with the lethality of this mutant, the DoHM mutant did not support DNA synthesis in an in vitro replication assay using purified replication proteins (*Figure 7A*; *Yeeles et al., 2015*).

To identify the step during replication initiation that was defective, we first assayed association of helicase-activation proteins with wild-type and the DoHM mutant proteins. We used bead-attached templates to monitor association of Cdc45, GINS, and Mcm10 with origin DNA associated with the indicated Mcm2-7 (*Lõoke et al., 2017*). Although there is less association of these proteins with the DoHM mutant, the reduction corresponds with the reduced loading of this version of Mcm2-7 (*Figures 4C* and *7C*). Thus, DDK-dependent association of Cdc45, GINS and Mcm10 with Mcm2-7-associated DNA was not compromised in the context of DoHM Mcm2-7 single hexamers (*Figure 7B,C*).

We also addressed the impact of the DoHM mutant on origin DNA unwinding using an assay that detects the formation of supercoiled DNA as a consequence of the unwinding process (*Douglas et al., 2018*). We initially used a 3.8 kb circular template that allows detection of extensive DNA unwinding. Consistent with the lack of replication, the DoHM mutant showed no evidence of DNA unwinding (*Figure 7D*) in this assay, as indicated by the lack of formation of supercoiled DNA. We performed the same assay with a ~ 600 bp circle that allows detection of small topological changes that are formed after CMG formation but do not require Mcm10 action (*Douglas et al., 2018*). Consistent with productive Cdc45 and GINS association with the DoHM mutant, we observed equivalent amounts of these initial intermediates for wild-type and DoHM proteins (*Figure 7E*). Both the small topological changes observed with the DoHM mutant and the finding that these intermediates are formed in the presence or absence of Mcm10 indicated that they are the result of initial DNA melting. Together, these findings indicate that CMG formation and initial DNA melting/distortion are independent of double-hexamer formation but more extensive origin DNA unwinding requires these interactions.

## Discussion

Our findings identified an essential and highly-conserved motif in Mcm4 that is required for stable double-hexamer formation but not initial loading of Mcm2-7. Using single-molecule studies, we

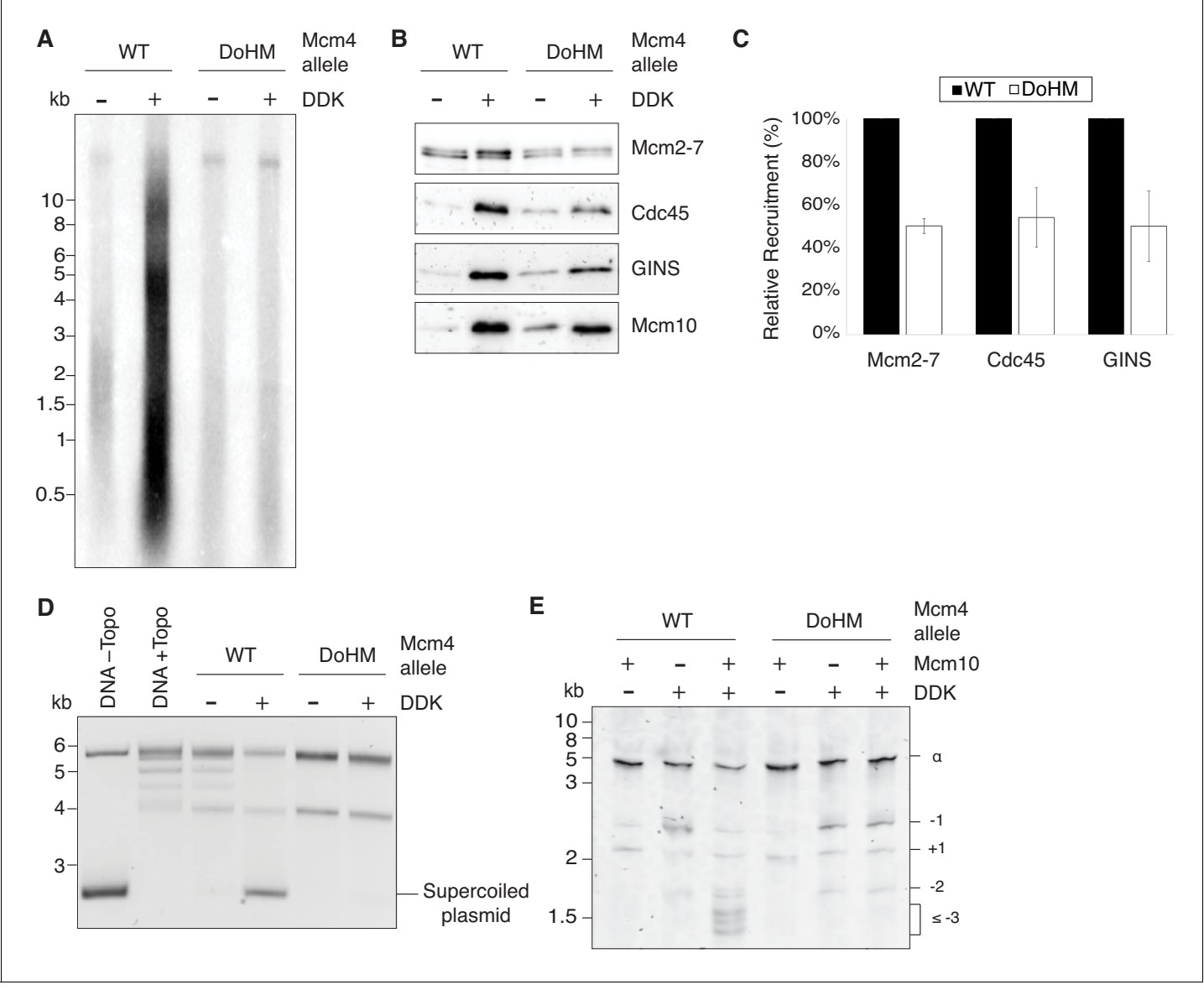

**Figure 7.** Double-hexamer formation is essential for DNA replication initiation. (**A**) The DoHM mutant is defective in DNA replication. The Mcm2-7 complexes were tested in a reconstituted replication assay. The helicase was loaded onto DNA templates and subsequently phosphorylated by DDK. Replication proteins, dNTPs and [$\alpha$-$^{32}$P]-dCTP were added to initiate DNA replication, and the replication products were detected by phosphor imaging. Minus-DDK controls show initiation of DNA replication is dependent on DDK phosphorylation. (**B**) The DoHM mutant can recruit Cdc45, GINS, and Mcm10. Mcm2-7 complexes were tested for CMG formation. The helicase was loaded onto bead-coupled DNA template and subsequently phosphorylated by DDK. The previous reaction mix was removed prior to addition of helicase-activation proteins. Bead-associated proteins were washed with high-salt buffer and detected by immunoblot. Omission of DDK was used as a control for non-specific DNA binding. (Mcm2-7 loading is DDK-independent). (**C**) The DoHM mutant defect for Cdc45 and GINS recruitment is correlated with the reduced helicase loading observed for this form of Mcm2-7. Data from three independent CMG-formation experiments (including that shown in B) were quantified using ImageJ. Error bars indicate the SD. For each Cdc45 and GINS measurement, the -DDK signal was subtracted from the corresponding +DDK signal. (**D**) The DoHM mutant is defective for extensive DNA unwinding. Mcm2-7 was loaded onto Topo I-relaxed 3.8 kb plasmids followed by addition of S-CDK, DDK, and helicase-activation proteins to stimulate CMG formation and activation. The reaction was quenched by SDS, separated on an agarose gel, and DNA was detected by ethidium bromide staining. DNA unwinding is detected by the formation of supercoiled DNA and is dependent on DDK phosphorylation. (**E**) The DoHM mutant allows initial topological changes that occur after CMG formation. Reactions were performed as in (**D**) except Mcm2-7 complexes were loaded onto a 616 bp circle and unwinding products were detected by SYBR Gold staining. Control reactions lacking DDK or Mcm10 are indicated. Identity of topoisomers is indicated on the right (see *Figure 7—figure supplement 1*).

DOI: https://doi.org/10.7554/eLife.45538.017

The following source data and figure supplement are available for figure 7:

*Figure 7 continued on next page*

*Figure 7 continued*

**Source data 1.** Primary data and quantification for graph in *Figure 7C*.
DOI: https://doi.org/10.7554/eLife.45538.019
**Figure supplement 1.** Assignment of relative supercoiling states for *Figure 7E*.
DOI: https://doi.org/10.7554/eLife.45538.018

show that this mutant does not prevent initial interactions between the two Mcm2-7 hexamers but that the resulting interactions are incomplete and unstable. Interestingly, these analyses reveal evidence consistent with a limited window of opportunity for double-hexamer formation after recruitment of the second Mcm2-7. Finally, using a DoHM mutant that prevents stable double-hexamer formation, we demonstrate that the Mcm2-7 double hexamer is required for extensive origin DNA unwinding but not initial recruitment of helicase-activating proteins or origin melting.

## Double-hexamer formation is a facilitated event

There are two general models to explain how two Mcm2-7 complexes form a double hexamer. The first model proposes that the two Mcm2-7 helicases are loaded independently, then slide along DNA, and form a double hexamer through interactions between their N-terminal domains independent of other proteins (*Coster and Diffley, 2017*). An alternative model is that helicase-loading proteins are required to facilitate both loading of the Mcm2-7 complexes and double-hexamer formation. Consistent with the latter model, kinetic analyses of the single-molecule studies presented here provide evidence that double-hexamer formation is facilitated. For both wild-type and the DoHM mutant, the formation of FRET between Mcm7 subunits after arrival of the second Mcm2-7 is rapid ($k_{DHex}$ = 0.13–0.15 s$^{-1}$; *Figures 5D–i, E–i, 6A and B*). Indeed, all double-hexamer formation observed occurs within 30 s after arrival of the second Mcm2-7. In addition, double hexamer interaction is detected significantly sooner (1/0.15 s$^{-1}$ = ~7 s) after second Mcm2-7 arrival than second Mcm2-7 ring closure (~57 s, *Ticau et al., 2017*), indicating that interactions between the hexamers anticipates completion of loading of the second Mcm2-7. These comparisons argue against a model in which helicase loading for both hexamers is complete at the time hexamer-hexamer interactions first occur. Instead, our findings strongly suggest that loading of the second hexamer and double-hexamer formation are coordinated.

We propose that double-hexamer formation requires one or more helicase-loading proteins and that the limited dwell times of these proteins on DNA results in a short window of opportunity for double-hexamer formation. This hypothesis is consistent with the observation that, in both wild-type Mcm2-7 and the DoHM mutant cases, two Mcm2-7 complexes either rapidly transition to a high-FRET state after the second Mcm2-7 arrives or do not visit the high-FRET state at all (*Figure 6*). Similarly, neither the rare wild-type nor frequent DoHM mutant instances that show a high-FRET to low-FRET transition ever displayed a second interval of high FRET (0 of 3, *Figures 6C* and 0 of 75, *Figure 6D*). These observations suggest that the establishment of the hexamer-hexamer interactions resulting in high FRET are associated with an irreversible step that prevents subsequent high-FRET transitions if that initial complex is unstable. If two Mcm2-7 complexes could form a double-hexamer unassisted, we would expect subsequent transitions into the high-FRET state in these cases, as the two non-interacting Mcm2-7 complexes frequently remain on the DNA for >100 s. The long dwell times of the single hexamers also argues against models in which the instability of single Mcm2-7 complexes on DNA prevents a second attempt to form the double hexamer. The hypothesis that helicase-loading proteins facilitate double-hexamer formation is consistent with previous single-molecule studies showing that Cdc6, Cdt1 and ORC are all released from the DNA shortly after arrival of the second Mcm2-7 (*Ticau et al., 2015*). Finally, such a mechanism would have the advantage of only allowing double-hexamer formation during loading and preventing such interactions from occurring at other times of the cell cycle (e.g. as replication forks converge).

Although release of any of the helicase-loading factors could prevent second attempts to form a double hexamer, the average release time of the second Cdc6 after arrival of the second Mcm2-7 (~23 s) would best fit the window of opportunity that we observe (i.e. DH are not formed longer than 30 s after second Mcm2-7 arrival). Another possibility is that closure of the Mcm2-7 ring terminates the ability of Mcm2-7 to form a double hexamer. However, the average time for this event relative to arrival of the second Mcm2-7 (~57 s) is much longer than the average time for double-

hexamer formation after the second Mcm2-7 arrives, making release of Cdc6 a more likely candidate for the process that prevents subsequent double-hexamer formation.

We did not see a dramatic difference in the number of second Mcm2-7 association events for the DoHM mutant, suggesting that the interactions interrupted by this mutant are not required to recruit the second Mcm2-7 to the DNA. This is consistent with models in which the recruitment of two Mcm2-7 complexes is independent of Mcm-Mcm interactions (*Coster and Diffley, 2017*). We note, however, that, under the conditions of our single-molecule reactions, only one ORC molecule is required for these two events (*Ticau et al., 2015*). It remains possible that the initial recruitment of the second Mcm2-7 involves Mcm-Mcm interactions that are not detected by the FRET probe or inhibited by the DoHM mutant. Development of additional modified Mcm2-7 complexes to detect interactions at other sites will help to address this possibility.

Although there is no difference in the number of second Mcm2-7 association for the DoHM mutant, we observed a shorter lifetime of the association of two DoHM mutant complexes with DNA (average 140 ± 9 s, *Figure 6D*) compared to wild-type Mcm2-7 (average 288 ± 23 s, *Figure 6C*). The most likely explanation of this difference is that Mcm2-7 single hexamers have a reduced stability on DNA relative to a Mcm2-7 double hexamer. Although previous studies showed that single Mcm2-7 hexamers can be loaded onto the DNA in a salt-stable manner (*Ticau et al., 2015*), it is possible that, the single DoHM mutant complexes are more prone to dissociate from the DNA. In support of this hypothesis, we see the same shorter dwell times for pairs of wild-type Mcm2-7 that fail to form a double-hexamer (*Figure 6C*, blue lines). This is consistent with the decreased helicase loading of the DoHM mutant in the ensemble loading assay (*Figure 4C*). Although it is also possible that the decreased loading is due to single hexamers sliding off the of the linear DNA substrate used in the initial bulk assays (e.g *Figures 2–4*), this is unlikely because we see the same reduction in DoHM loading when a circular DNA template is used (*Figure 7B,C*).

We note that in both the wild-type and DoHM mutant experiments there are frequent instances when two Mcm2-7 complexes associate with the DNA for long periods of time but show no transitions to high FRET (*Figure 6C and D*, non-DH traces). As discussed above, one possibility is that these molecules have lost a helicase-loading protein(s) required for double-hexamer formation before establishing this complex. Alternatively, the lack of double-hexamer formation in these cases could be because the two Mcm2-7 complexes are loaded on the DNA in an incorrect orientation. For example, two sequential 'first' loading events on the same DNA would result in two Mcm2-7 complexes in the same orientation. We note that there are many instances (primarily for the DoHM mutant) in which initial hexamer-hexamer interactions are unstable, but the separated hexamers do not make a second attempt to form a stable double hexamer. In these cases, the second hypothesis cannot explain the lack of a second attempt to form the double hexamer as the two Mcm2-7 complexes must have been in the appropriate orientation to form the initial high-FRET (pseudo-DH) interaction.

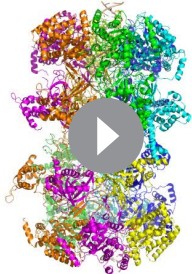

**Video 1.** Illustration of the DoHM interactions in the Mcm2-7 double hexamer (*Abid Ali et al., 2017*). The DoHM of Mcm4 (green), the region of Mcm7 that interacts with the DoHM (light blue), and the region of Mcm5 from the opposite hexamer that interacts with the DoHM (yellow) are all shown in space filling representation. The remainder of the structure is shown as a ribbon structure with Mcm6 in orange, Mcm2 in magenta, and Mcm3 in dark blue. Note that there are two DoHM-involved interfaces that are located on opposite sides of the Mcm2-7 double hexamer.
DOI: https://doi.org/10.7554/eLife.45538.020

## Structure of double-hexamer and the DoHM interactions

The lower $E_{FRET}$ value of the mutant pseudo-DH state suggests that the DoHM mutation interferes with proper interactions at the double-hexamer interface. This hypothesis is consistent with the cryo-EM structures of the Mcm2-7 double hexamer (*Abid Ali et al., 2017*; *Li et al., 2015*; *Noguchi et al., 2017*). In each of the structures, the DoHM is part of a key double-hexamer interface. The DoHM forms a loop that interacts with Mcm5 N-terminal domain from the opposite hexamer. In addition, the DoHM interacts with one end of an extended Mcm7 alpha helix from

the same hexamer. Importantly, the other end of this Mcm7 helix interacts with Mcm5 of the opposite hexamer (*Figure 4—figure supplement 2*), raising the possibility that the DoHM positions this helix to appropriately interact with the other hexamer. These interactions are present in the Mcm2-7 double-hexamer structures both with and without DNA (*Abid Ali et al., 2017*; *Li et al., 2015*; *Noguchi et al., 2017*). There are two sets of these three-subunit interactions (Mcm4-Mcm5-Mcm7) located on opposite sides of the double-hexamer interface (*Video 1*). Thus, although the DoHM is a small region compared to the large double-hexamer interface, disrupting this motif has the potential to interrupt the interactions involving 6 of the 12 subunits participating in the double-hexamer interface.

## The double hexamer is required for extensive DNA unwinding

In this study, we show that a double hexamer is not required to recruit helicase-activation proteins or perform initial DNA unwinding. Although it was assumed that double hexamers are an essential part of DNA replication initiation, the replication initiation step that required the double hexamer was unknown. The DoHM mutant allowed us to evaluate the role of the double-hexamer complex. We found that the DoHM mutant can assemble CMG complexes that perform limited initial DNA unwinding (*Figure 7*). This degree of topological change corresponds to the modest unwinding of less than one turn of DNA per helicase previously observed as a consequence of CMG formation (*Douglas et al., 2018*). The detection of this initial DNA unwinding indicates that the interactions of Cdc45 and GINS with the DoHM mutant are productive. Since the DoHM mutant does not form stable double hexamers, this initial unwinding is likely to arise from independent helicases. We note that based on the measured lifetimes of the DoHM mutant pseudo-double-hexamers, helicase-activation proteins are not in a position to stabilize the pseudo-double hexamers before they dissociate. In these experiments, DDK phosphorylation is performed for 20 min *prior* to addition of the helicase-activation proteins, which is much longer than the ~20 s (1/0.052 s$^{-1}$; *Figure 6B*) on average that the DoHM mutant is in the pseudo-DH state. We also note that once separated, we observe no evidence of subsequent hexamer-hexamer interactions. Thus, our findings strongly suggest that this initial unwinding occurs by manipulation of the DNA within a single CMG complex rather than through the coordinated action of two opposing helicases.

In contrast to the ability of the DoHM mutant to catalyze initial DNA unwinding, this form of the helicase cannot perform more extensive DNA unwinding that requires Mcm10. This result strongly suggests that double-hexamer interactions are required to drive this DNA unwinding. If true, this hypothesis would suggest that additional DNA unwinding outside the central channel of the helicase requires a tight interface between the two helicases. One way, this tight interface could be necessary is if opposing action of the two helicases drives DNA unwinding and formation of the replication bubble. A requirement for the double-hexamer could occur before the two hexamers separate or in a second event that involves the two hexamers coming back together after they separate during CMG formation. The latter hypothesis is consistent with recent studies suggesting that CMG formation causes double-hexamer separation followed by an Mcm10-dependent transition involving two CMG complexes (*Douglas et al., 2018*; *Langston and O'Donnell, 2019*). Taken together with previous studies, our studies support a model in which DNA unwinding during DNA replication initiation occurs in at least three stages: 1) initial limited DNA melting upon CMG formation, 2) further DNA unwinding that results in ssDNA-strand extrusion that requires double-hexamer interactions and Mcm10, and 3) extensive DNA unwinding that is mediated single CMG complexes translocating on ssDNA at replication forks.

We also considered whether the DNA unwinding defects observed for the DoHM mutant were due to Mcm10-binding or Mcm2-7-ATPase defects; however, other data argue against these explanations. First, Mcm2-7 ATPase activity is required for DNA unwinding beyond that detectable upon CMG formation, making an Mcm2-7 ATPase defect a plausible explanation for the defect observed. We note, however, that the DoHM mutant is functional for helicase loading, which also requires Mcm2-7 ATPase activity (*Coster et al., 2014*; *Kang et al., 2014*) making an ATPase defect unlikely. Second, although Mcm10 also is required for the transition from initial to more extensive unwinding, multiple findings argue against the DoHM mutant having a defect in Mcm10 binding. Although the N-terminal domain of Mcm4 has been observed to interact with Mcm10 in the absence of the rest of Mcm2-7 (*Quan et al., 2015*), in the context of the full complex, Mcm10's interactions with Mcm2 and Mcm6 are more important (*Douglas and Diffley, 2016*; *Lõoke et al., 2017*; *Mayle et al.*,

2019). Additionally, crosslinking studies of the CMG-Mcm10 structure showed only three Mcm10 crosslinking sites with Mcm4, all of which are distant from the DoHM mutation (*Mayle et al., 2019*). Most importantly, we found that the DoHM mutant maintains the ability to form high-salt-resistant interactions with Mcm10 (*Figure 7*).

The identification of mutations that prevent double-hexamer formation provides an important tool to further explore the role of this complex in replication initiation. Although we have provided a first look of the role of the double hexamer in initiation, there is more to be understood about how this intermediate facilitates protein associations and origin unwinding. In addition, although our studies provide strong evidence that the loading of the second Mcm2-7 is coordinated with formation of the double hexamer, the mechanism of this coordination is an important area for future investigation.

# Materials and methods

## Key resources table

| Reagent type (species) or resource | Designation | Source or reference | Identifiers | Additional information |
|---|---|---|---|---|
| Antibody | anti-Mcm2-7 | Bell Lab | UM174 | 1:10,000 (rabbit polyclonal) |
| Antibody | anti-Cdc45 | *Lõoke et al., 2017* | HM7135 | 1:2000 (rabbit polyclonal) |
| Antibody | anti-GINS | *Lõoke et al., 2017* | HM7128 | 1:2000 (rabbit polyclonal) |
| Antibody | anti-Mcm10 | *Lõoke et al., 2017* | HM7602 | 1:2000 (rabbit polyclonal) |
| Antibody | anti-cMyc agarose | Sigma | Sigma: A7470 | |
| Antibody | anti-V5 agarose | Sigma | Sigma: A7345 | |
| Chemical compound, drug | DY549-P1-maleimide | Dyomics | Dyomics: 549P1-03 | |
| Chemical compound, drug | DY649-P1-maleimide | Dyomics | Dyomics: 649P1-03 | |
| Chemical compound, drug | SYBR Gold nucleic acid stain | Invitrogen | Invitrogen: S11494 | |
| Commercial assay or kit | illustra MicroSpin G-50 columns | GE Healthcare | GE Healthcare: 27533001 | |
| Gene (*Saccharomyces cerevisiae*) | MCM2 | *Saccharomyces* Genome Database | SGD: S000000119 | |
| Gene (*Saccharomyces cerevisiae*) | MCM3 | *Saccharomyces* Genome Database | SGD: S000000758 | |
| Gene (*Saccharomyces cerevisiae*) | MCM4 | *Saccharomyces* Genome Database | SGD: S000006223 | |
| Gene (*Saccharomyces cerevisiae*) | MCM5 | *Saccharomyces* Genome Database | SGD: S000004264 | |
| Gene (*Saccharomyces cerevisiae*) | MCM6 | *Saccharomyces* Genome Database | SGD: S000003169 | |
| Gene (*Saccharomyces cerevisiae*) | MCM7 | *Saccharomyces* Genome Database | SGD: S000000406 | |
| Other | TransFluoSpheres Streptavidin-Labeled Microspheres | Thermo Fisher Scientific | Thermo Fisher: T10711 | |
| Other | Amersham Hybond-XL | GE Healthcare | GE Healthcare: RPN303S | Membrane for imaging replication assay |
| Peptide, recombinant protein | DNase I | Worthington | Worthington: LS006333 | |

*Continued on next page*

*Continued*

| Reagent type (species) or resource | Designation | Source or reference | Identifiers | Additional information |
|---|---|---|---|---|
| Peptide, recombinant protein | Proteinase K | Thermo Fisher Scientific | Thermo Fisher: AM2548 | |
| Peptide, recombinant protein | Nb.BsrDI | New England Biolabs | New England Biolabs: R0648S | |
| Peptide, recombinant protein | Peptide for coupling to dyes for Sortase labeling | *Ticau et al., 2015* | | NH2-CHHHHHHH HHHLPETG |
| Sequence-based reagent | oligo for 1.3 kb template | IDT | | 5'-Biotin-GATCGGTG CGGGCCTCTTCGC-3' |
| Sequence-based reagent | oligo for 1.3 kb unlabeled template | IDT | | 5'-GGAAAGCGGGCA GTGAGCGC-3' |
| Sequence-based reagent | oligo for 1.3 kb fluorescently-labeled template | IDT | | 5'-Alexa488-GGAAAGC GGGCAGTGAGCGC-3' |
| Software, algorithm | Matlab | Mathworks | | |

## Protein purifications

Wild-type Mcm2–7/Cdt1 and ORC complexes were purified as described previously (*Kang et al., 2014*). Wild-type Cdc6 was purified as described in *Frigola et al. (2013)*. DDK, S-CDK, Sld3/7, Cdc45, Sld2, Dpb11, GINS, Mcm10, Polymerase epsilon, Polymerase alpha/primase, Polymerase delta, RPA, Ctf4, RFC, PCNA, Mrc1, Csm3-Tof1, and Topo II were purified as described in *Lõoke et al. (2017)*. Mutant Mcm2-7/Cdt1 complexes were purified as described in *Kang et al. (2014)* with the following modifications. For each Mcm2-7 mutant complex, the corresponding wild-type proteins were epitope-tagged with either c-Myc or V5. In the strains expressing the Mcm2 Δ2–177 and Mcm6 Δ2–105, the wild-type *MCM2* and *MCM6* genes were tagged with c-Myc, to allow the endogenous 13Myc-tagged Mcm2 or Mcm6 subunits to be depleted by incubating with anti-c-Myc agarose (Sigma) before applying the Mcm2-7 mutant complex to a Superdex 200 gel filtration column. In strains expressing mutant Mcm4 protein, the wild-type *MCM4* gene was tagged with V5. Mcm2-7 complexes containing the endogenous V5-tagged Mcm4 subunits were depleted by incubating with anti-V5 agarose (Sigma) before application to a Superdex 200 gel filtration column. Yeast strains and plasmids used are listed in *Supplementary file 1 - tables 2 and 3*, respectively.

## Helicase-loading and double-hexamer formation assays

Helicase-loading and double-hexamer formation assays were performed as described in *Kang et al. (2014)*.

## Fluorescent labeling of wild-type Mcm2-7/Cdt1 and Mcm2-7$^{4-178A}$/Cdt1

SORT-tagged wild-type Mcm2-7/Cdt1 was purified and labeled with either DY549-P1 or DY649-P1 (Dyomics) as described in *Ticau et al. (2015)*. SORT-tagged Mcm2-7$^{4-178A}$/Cdt1 was purified and labeled using the same protocol with the following modifications. In the strain expressing the Mcm4-178A, the wild-type *MCM4* gene was tagged with V5 to allow the endogenous V5-tagged Mcm4 subunits to be depleted by incubating with anti-V5 agarose (Sigma) before dye coupling as described above.

## Determining fractional labeling of Mcm2-7

To determine what fraction of Mcm2-7 molecules were fluorescently labeled, 20 ml of DY549-SORT labeled Mcm2-7 was mixed with maleimide-DY-649P1 dissolved in anhydrous DMSO, in a 1:1 molar ratio at 4℃ for 10 min. The reaction was terminated with 2 mM DTT. The double-labeled Mcm2-7 (10 nM) was added to a slide coupled to origin DNA, 0.5 nM of purified ORC, 2 nM of purified Cdc6 and monitored Mcm2-7-DNA colocalization (to ensure that we were monitoring fully assembled complexes). The fraction of maleimide-DY-649P1-labeled Mcm2-7 molecules that also contained DY-549P1 was determined and reported as the percent labeling by the DY-549P1 (we assume that

coupling of maleimide-DY-649P1 to Mcm2-7 is not influenced by the presence or absence of the 549 label). The same protocol was used for DY649-SORT labeled Mcm2-7, but this complex was double-labeled with maleimide-DY-549P1.

## Single-molecule microscopy and FRET data analysis procedure

Single molecule experiments were performed as described in *Ticau et al. (2015)* for monitoring FRET with the following modifications. TransFluoSpheres Streptavidin-Labeled Microspheres (ThermoFisher) were added to the slide prior to DNA attachment to facilitate tracking of stage drift during the course of the experiment. Helicase-loading reactions contained 0.5 nM ORC, 2 nM Cdc6, 5 nM DY549-SORT labeled Cdt1/Mcm2–7, and 5 nM DY649-SORT labeled Cdt1/Mcm2–7. During image acquisition, a computer-controlled focus adjustment (using a 785 nm laser) was applied continuously over the course of the experiment (~20 min). FRET data analysis was performed as described in *Ticau et al. (2015)* except that the fluorescent microspheres were used for drift correction. Spots with DNA colocalization of one donor-fluorophore and one acceptor-fluorophore labeled were manually selected for $E_{FRET}$ calculation. To calculate apparent FRET efficiencies, each fluorescence intensity trace was background subtracted using custom Matlab (MATHWORKS) image processing software that has been previously described (*Friedman and Gelles, 2012*). For each trace, baseline segments (trace intervals lacking any spot) were joined after smoothing with a low-pass filter, and that smoothed baseline was subtracted from the initial fluorescence trace. FRET efficiency was calculated using $E_{FRET} = I_{Acceptor}/(I_{Acceptor} + I_{Donor})$ where $I_{Acceptor}$ and $I_{Donor}$ are the acceptor and donor emission intensities observed during donor excitation, respectively. The two-dimensional Gaussian kernel histograms and the one-dimensional histogram fits in *Figure 5* were generated using code from *Gelles (2019)* (copy archived at https://github.com/elifesciences-publications/jganalyze). Two-dimensional histograms used bandwidths 5 s and 0.05 on the time and $E_{FRET}$ axes, respectively and were normalized so that the probability density in each 2.67 s time slice integrated to one.

## Kinetic analysis

Pooled single-molecule fluorescence records from helicase loading experiments from Mcm2-7 or Mcm2-7[4-178A] were globally fit to generalized three-state kinetic models (i.e. all possible inter-state transitions were allowed). Fitting was performed using coupled hidden Markov models using an empirical Bayesian approach to estimate priors (*van de Meent et al., 2014*) as implemented in program ebfret-gui (https://github.com/ebfret/ebfret-gui/commit/28e548ace84190c91c4ca354f41e-fa5952a7895f). Outlier points with $E_{FRET}$ outside the range [−0.25, 0.85] (3.5% of Mcm2-7 data and 7% of Mcm2-7[4-178A] data) were excluded from the analysis as were individual DNA molecule records (7 Mcm2-7 and 11 Mcm2-7[4-178A]) with anomalously low signal-to-noise as judged by their containing more than 10 outliers each. Default priors were used except that the state $E_{FRET}$ values were strongly constrained to those from independent fits (in *Figure 5* and *Supplementary file 1 - table 1*) by setting the (hyper)parameters μ and β (*van de Meent et al., 2014*) to (0.06, 0.06, 0.57) and (1,000, 1,000, 1,000) for Mcm2-7 and (0.08, 0.08, 0.47) and (10,000, 10,000, 10,000) for Mcm[4-178A].

## CMG formation assay

The DNA plasmid template pUC19-*ARS1* was randomly biotinylated and coupled to streptavidin-coated magnetic beads as described previously (*Heller et al., 2011*). Each incubation step was performed in a thermomixer (Eppendorf) with shaking at 1250 rpm at 25˚C. Mcm2-7 loading was performed by incubating 0.48 pmol of ORC, 0.52 pmol of Cdc6, and 1.14 pmol of Mcm2–7/Cdt1 with 0.125 pmol template DNA in 25 mM HEPES-KOH (pH 7.6), 10 mM magnesium acetate, 225 mM potassium glutamate, 2 mM DTT, 0.02% NP-40, 5% glycerol, 5 mM ATP, 20 mM phosphocreatine, and 0.2 μg of creatine kinase for a total volume of 10 μL. Reactions were incubated for 20 min, at which point 1.3 pmol of DDK was added and incubation was continued for a further 20 min. The supernatant was then removed by applying the reaction to a DynaMag-2 magnet (ThermoFisher Scientific) to isolate the DNA coupled to magnetic streptavidin beads from the supernatant. CMG formation was then initiated by adding 20 μL of 0.6 pmol CDK, 1 pmol Sld3/7, 1 pmol Cdc45, 1.24 pmol Sld2, 0.8 pmol Dpb11, 5 pmol GINS, 0.06 pmol Mcm10, 1.05 pmol RPA, and 0.6 pmol Pol ε in 25 mM HEPES-KOH (pH 7.6), 10 mM magnesium acetate, 250 mM potassium glutamate, 1 mM DTT,

0.02% NP-40, 8% glycerol, 5 mM ATP, and 0.4 mg/ml BSA and incubated for 30 min. Reactions were washed with 300 mM potassium chloride, 25 mM HEPES-KOH (pH 7.6), 5 mM magnesium acetate, 10% glycerol, and 0.01% NP-40 three times. Proteins were released from the DNA by incubating with 5 U of DNase I (Worthington) in 10 µL of 25 mM HEPES-KOH (pH 7.6), 5 mM magnesium acetate, 200 mM sodium chloride, 5% glycerol, 0.02% NP-40, and 2 mM calcium chloride for 20 min at 25°C before immunoblotting.

## Solution replication assay

Each incubation step was performed in a thermomixer (Eppendorf) with shaking at 1250 rpm at 25°C. Mcm2-7 loading and DDK phosphorylation was performed in the same conditions as the CMG formation assay except with a soluble 11.9 kb pUC19-ARS1 plasmid template. After DDK phosphorylation, replication was initiated by adding 20 µL of 0.6 pmol CDK, 1 pmol Sld3/7, 2.6 pmol Cdc45, 1.24 pmol Sld2, 0.8 pmol Dpb11, 5 pmol GINS, 0.02 pmol Mcm10, 0.6 pmol Pol ε, 1.5 pmol Pol α, 0.5 pmol Topo II, 0.6 pmol Ctf4, 2.32 pmol RPA, 0.5 pmol RFC, 0.4 pmol PCNA, 0.5 pmol Mrc1, 0.6 pmol Csm3/Tof1, 0.6 pmol Pol δ in 12.5 mM HEPES-KOH (pH 7.6), 5 mM magnesium acetate, 125 mM potassium glutamate, 1 mM DTT, 0.01% NP-40, 4% glycerol, 1.5 mM ATP, 10 mM phosphocreatine, 3 µg of creatine kinase, 0.2 mg/ml BSA, 100 µM rNTP, 10 µM dNTP, and 10 µCi [α-P$^{32}$]dCTP directly to the reaction. Following 60 min of incubation, reactions were quenched with 30 µL of 50 mM EDTA. Unincorporated nucleotides were removed with Illustra MicroSpin G-50 columns (GE Healthcare), and samples were separated on a 0.6% alkaline agarose gel in 30 mM sodium hydroxide, 2 mM EDTA. Gels were dried onto Amersham Hybond-XL (GE Healthcare) and imaged using a phosphor screen. Gels were scanned using a Typhoon phosphorimager (GE Healthcare).

## DNA unwinding assays

For DNA unwinding assays using the 3.8 kb template, 25 fmol soluble 3.8 kb pUC19-ARS1 plasmid template was relaxed with 0.4 pmol Topo I for 30 min. Each incubation step was performed in a thermomixer (Eppendorf) with shaking at 1250 rpm at 25°C unless otherwise indicated. Twenty-five minutes of Mcm2-7 loading and 30 min of DDK phosphorylation were performed in the same conditions as the CMG formation assay. DNA unwinding was then initiated using the same protein concentrations and final buffer concentrations as the CMG formation assay with the addition of 0.4 pmol of Topo I. This mix was added directly to the reaction. After 40 min, the reaction was quenched with 13 mM EDTA, 0.3% SDS, and 0.1 mg/ml Proteinase K and incubated for 20 min at 42°C with shaking at 1250 rpm. Samples were extracted with phenol:chloroform:isoamylalcohol (25:24:1), ethanol precipitated, and the DNA pellet was resuspended in 1x Tris-EDTA. Samples were run on native 1.5% agarose TAE gels at 1.5 V/cm for 17 hr. Gels were stained with ethidium bromide for 30 min and destained with milliQ-purified water (Millipore Sigma) for 1 hr before imaging.

DNA unwinding of 616 bp template was performed similarly to DNA unwinding with the 3.8 kb template but with the following modifications. 616 bp pUC19-ARS1 closed-circular template was made following the protocol in *Douglas et al. (2018)* except the DNA was not radiolabeled. At the end of the reaction, samples were run on native 3.5% 29:1 acrylamide:bis-acrylamide 1x TBE gels at 3 V/cm at 4°C for 20 hr. Gels were stained with SYBR Gold (Invitrogen) for 30 min before imaging.

## Assigning relative supercoiling states

Experiment was performed similar to *Douglas et al. (2018)*. Briefly, 6 fmol/µl 616 bp DNA was nicked with 0.25 U/µl Nb.BsrDI enzyme (NEB) for 1 hr at 65°C. DNA was then extracted with phenol:chloroform:isoamylalcohol (25:24:1), ethanol precipitated, and the DNA pellet was resuspended in water. 25 fmol of the nicked DNA was incubated in the indicated ethidium bromide concentrations for 1 hr at room temperature then ligated at 18°C overnight with 10 u/µl T4 DNA ligase (NEB). DNA was phenol:chloroform extracted, ethanol precipitated, and resuspended in 1 × Tris EDTA before analysis by electrophoresis, as described above. See *Figure 7—figure supplement 1*.

## Acknowledgements

We are grateful to Shalini Gupta and Alexandra Pike for comments on the manuscript, Mike Maloney for initial analysis of DoHM mutants in replication assays, and Yingwu Xu and Christian Ramsoomair

for preparation of a subset of the proteins required for the biochemical assays used in this manuscript.

## Additional information

### Funding

| Funder | Grant reference number | Author |
|---|---|---|
| National Institute of General Medical Sciences | GM52339 | Stephen P Bell |
| Howard Hughes Medical Institute | Investigator | Stephen P Bell |
| G. Harold and Leila Y. Mathers Foundation | | Jeff Gelles |
| National Cancer Institute | P30-CA14051 | Stephen P Bell |
| National Institute of General Medical Sciences | GM81648 | Jeff Gelles |

The funders had no role in study design, data collection and interpretation, or the decision to submit the work for publication.

### Author contributions

Kanokwan Champasa, Resources, Data curation, Formal analysis, Validation, Investigation, Methodology, Writing—original draft; Caitlin Blank, Resources, Formal analysis, Validation, Investigation, Methodology; Larry J Friedman, Data curation, Software, Formal analysis, Supervision, Validation, Methodology, Writing—review and editing; Jeff Gelles, Conceptualization, Data curation, Software, Formal analysis, Supervision, Funding acquisition, Validation, Visualization, Methodology, Project administration, Writing—review and editing; Stephen P Bell, Conceptualization, Supervision, Funding acquisition, Validation, Methodology, Project administration, Writing—review and editing

### Author ORCIDs

Larry J Friedman (iD) http://orcid.org/0000-0003-4946-8731
Jeff Gelles (iD) https://orcid.org/0000-0001-7910-3421
Stephen P Bell (iD) https://orcid.org/0000-0002-2876-610X

### Decision letter and Author response

Decision letter https://doi.org/10.7554/eLife.45538.026
Author response https://doi.org/10.7554/eLife.45538.027

## Additional files

### Supplementary files

• Supplementary file 1. This file contains supplementary table 1 (fit parameters for EFRET distributions), supplementary table 2 (genotypes of yeast and bacterial expression strains used in this study), and supplementary table 3 (plasmids used in this study).
DOI: https://doi.org/10.7554/eLife.45538.021

• Transparent reporting form
DOI: https://doi.org/10.7554/eLife.45538.022

### Data availability

Source data for the plots in Figs. 3, 4, and 7 have been provided. Source data for the single-molecule experiments is provided as "intervals" files that can be read and manipulated by the Matlab program imscroll, which is publicly available: https://github.com/gelles-brandeis/CoSMoS_Analysis. The source data are archived as doi:10.5281/zenodo.2556799. All remaining data generated or analyzed during this study are included in the manuscript.

The following dataset was generated:

| Author(s) | Year | Dataset title | Dataset URL | Database and Identifier |
|---|---|---|---|---|
| Kanokwan Champasa | 2019 | Single-molecule source data files | https://zenodo.org/record/2556799 | Zenodo, 10.5281/zenodo.2556799 |

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
