## [Decision Letter]

Thank you for submitting your article "A conserved Mcm4 motif is required for Mcm2-7 double-hexamer formation and origin DNA unwinding" for consideration by *eLife*. Your article has been reviewed by three peer reviewers, including Michael R Botchan as the Reviewing Editor and Reviewer #1, and the evaluation has been overseen by a Reviewing Editor and Kevin Struhl as the Senior Editor. The following individual involved in review of your submission has agreed to reveal their identity: Alessandro Costa (Reviewer #2).

The reviewers have discussed the reviews with one another and the Reviewing Editor has drafted this decision to help you prepare a revised submission. The summary of the complied reviews are below.

Summary:

Cells prepare for replication initiation in G1, when they assemble a pre-replication complex (pre-RC) at each origin of replication. The pre-RC is assembled by ORC, Cdc6, and Cdt1 and consists of two head-to-head copies of the Mcm2-7 ATPase. Bell and colleagues identify a short, highly conserved Double Hexamer Motif (DoHM) in Mcm4 that is required for the formation of an Mcm2-7 double hexamer at origins of replication. In its absence, stable double hexamers fail to form (as measured by gel filtration after release of proteins from origin DNA). Based on single molecule analysis, the mutant MCM complexes form a pseudo double hexamer on DNA (less tightly interacting based in lower FRET signal) at almost normal rates, but these rapidly dissociate into monomers that remain on DNA but appear rarely transition back to the double hexamer state. Strikingly, the DoHM mutant complex is able to recruit Cdc45, GINS, and Mcm10 yet it is inactive for DNA unwinding, replication, and it does not support cell viability. The Mcm4 (DoHM), is thus dispensable for stable MCM loading onto DNA but essential for double hexamer formation. In particular, single-molecule analysis reveals that the MCM can form an initial interaction between two hexamers, but the complex falls apart. Intriguingly, a short time window is identified for double hexamer formation once the second MCM associates with the origin, suggesting that there is a special interaction between the first and second ring during the recruitment of the second hexamer, which allows head-to-head double hexamer engagement. Intriguingly, double hexamer formation is dispensable for the recruitment of Cdc45, GINS and Mcm10, but essential for DNA unwinding.

The other aspect of the paper concerns the models over how pre-RC assembly unfolds. Bell and colleagues previously used single molecule imaging to monitor pre-RC assembly and concluded that this process involves a single ORC complex bound to DNA that promotes two rounds of MCM loading without dissociating from DNA (Ticau et al., 2015). This model implies that the same ORC complex loads successive MCMs in opposite orientations (to achieve formation of a head-to-head MCM double hexamer). Subsequently, Coster and Diffley (2017) proposed an alternative model in which a second ORC complex binds to a cryptic ORC binding site in an opposite orientation relative to the first ORC and thereby recruit a second Mcm2-7 complex in the appropriate orientation, whereupon the two MCM complexes slide towards each other, forming the double hexamer. The authors of this paper interpret their data in terms of the Bell model. However, all the arguments are indirect. To better distinguish between the models, they might perform experiments that are explicitly designed for this purpose. For example, the single molecule analysis should be repeated on a model ARS DNA sequence designed to eliminate all possible cryptic ORC binding sites. If the kinetics of MCM loading is unaltered, this would provide powerful evidence against the Diffley model. As it stands, one cannot rule out the possibility that ORC dissociates and reassociates with a cryptic ORC site without leaving the diffraction limited spot. After discussions the readers concluded that this particular experiment was not required, but rather a more explicit and full discussion of the possibilities might be included in the Discussion section.

Essential revisions:

1) A very interesting aspect of the paper is discriminating between CMG formation and DNA unwinding. In a recent study, Diffley and colleagues have shown that stable CMG formation leads to DNA untwisting prior to replication-fork establishment. Can the authors comment on any change in DNA topology that occurs upon CMG formation prior to Mcm10 recruitment, using wild type vs. mutant Mcm4? One does realize that this experiment is technically challenging, however, it is feasible within the time possible for a revision. One of the reviewers puts the point this way:

“This paper is important because it reports the first discrete MCM mutant that disrupts double hexamer formation and therefore provides insight into the function of this structure. However, the analysis is incomplete. Given that the DoHM mutant is able to recruit Cdc45, GINS, and Mcm10, it is very surprising that it is inactive for DNA unwinding. Why is this so? To address this important gap in the data, the authors should use the higher resolution origin unwinding assay (Figure 2 in Douglas et al., 2018) to determine whether the Cdc45/GINS-induced and Mcm10-induced topological changes take place within this mutant. If one of these steps is blocked, the paper will represent a very impactful message about the role of double hexamer formation in replication initiation.”

2) Single-molecule data suggests that the second MCM hexamer is loaded and shows a long-lived association with DNA that cannot be distinguished from the first hexamer. The biochemical data however consistently show that modifying the DoHM in Mcm4 reduces MCM loading to 50%. Do the authors observe that second MCM recruitment leads to failure to form a long-lived DNA association more frequently in the mutant than in wild type? From the single-molecule data shown, there seems to be no difference in MCM loading, but only a difference in hexamer-hexamer association that leads to double hexamer formation. This does not seem to explain the 50% decrease in helicase loading. It would be good to clarify this point.

---

## [Author Response]

Essential revisions:1) A very interesting aspect of the paper is discriminating between CMG formation and DNA unwinding. In a recent study, Diffley and colleagues have shown that stable CMG formation leads to DNA untwisting prior to replication-fork establishment. Can the authors comment on any change in DNA topology that occurs upon CMG formation prior to Mcm10 recruitment, using wild type vs. mutant Mcm4? One does realize that this experiment is technically challenging, however, it is feasible within the time possible for a revision. One of the reviewers puts the point this way:“This paper is important because it reports the first discrete MCM mutant that disrupts double hexamer formation and therefore provides insight into the function of this structure. However, the analysis is incomplete. Given that the DoHM mutant is able to recruit Cdc45, GINS, and Mcm10, it is very surprising that it is inactive for DNA unwinding. Why is this so? To address this important gap in the data, the authors should use the higher resolution origin unwinding assay (Figure 2 in Douglas et al., 2018) to determine whether the Cdc45/GINS-induced and Mcm10-induced topological changes take place within this mutant. If one of these steps is blocked, the paper will represent a very impactful message about the role of double hexamer formation in replication initiation.”

We thank the reviewers for this important suggestion. We have performed the suggested experiment and have obtained a very interesting result. We find that the DoHM mutant can perform initial DNA melting but cannot (as before) perform more extensive DNA unwinding necessary for replication. This result clearly demonstrates that the associations of Cdc45 and GINS with the DoHM mutant are productive and more precisely defines the step during replication initiation that requires the Mcm2-7 double hexamer interactions as the transition between initial DNA melting and extensive origin DNA unwinding. In addition, by demonstrating that single CMG complexes can perform this initial DNA melting, these findings strongly suggest that the mechanism of initial DNA melting is likely to be mediated by interactions of Mcm2-7 with the encircled DNA but that the initiation of extensive unwinding is likely to require coordinated function of two Mcm2-7 complexes. Interestingly, because it has been proposed that the double-hexamer separates upon CMG formation, this finding suggests that either there is a double hexamer-dependent event that is required for extensive unwinding that occurs prior to CMG formation, or that there are important double-hexamer interactions that occur after initial separation. The latter hypothesis is consistent with recent studies from the O’Donnell lab suggesting that two CMG complexes come together to drive DNA unwinding. We have updated the paper throughout (Abstract, Results section and Discussion section) to emphasize this important finding.

To incorporate these findings we have added data to Figure 7E, added Figure 7—figure supplement 1, revised the Results section and the Discussion section.

2) Single-molecule data suggests that the second MCM hexamer is loaded and shows a long-lived association with DNA that cannot be distinguished from the first hexamer. The biochemical data however consistently show that modifying the DoHM in Mcm4 reduces MCM loading to 50%. Do the authors observe that second MCM recruitment leads to failure to form a long-lived DNA association more frequently in the mutant than in wild type? From the single-molecule data shown, there seems to be no difference in MCM loading, but only a difference in hexamer-hexamer association that leads to double hexamer formation. This does not seem to explain the 50% decrease in helicase loading. It would be good to clarify this point.

Although there is not a difference in the number of loading events, the dwell time of the molecules in the wild-type versus the DoHM mutants are significantly different. Because the bulk assays are endpoint assays, we believe that the 50% decrease is due to the different stability of single-hexamers versus double-hexamers on the DNA. Although it is possible that a difference caused by the DoHM mutant beyond the defect in double-hexamer formation contributes to this reduction, we note that we see very similar DNA dwell times for two wild-type Mcm2-7 complexes that do not form a double hexamer (see Figure 6C vs. 6D, blue lines). We have added a paragraph in the Results section and the Discussion section addressing this point.